# SP2: A Second Order Stochastic Polyak Method

Shuang Li [1], William J. Swartworth [1], Martin Takáč[2], Deanna Needell[1], and Robert M. Gower[3]

[1]Department of Mathematics, University of California, Los Angeles, USA
[2]Mohamed bin Zayed University of Artificial Intelligence, Abu Dhabi, UAE
[3]Center for Computational Mathematics, Flatiron Institute, Simons Foundation, New York, USA

## ABSTRACT

Recently the SP (Stochastic Polyak step size) method has emerged as a competitive adaptive method for setting the step sizes of SGD. SP can be interpreted as a method specialized to interpolated models, since it solves the *interpolation equations*. SP solves these equation by using local linearizations of the model. We take a step further and develop a method for solving the interpolation equations that uses the local second-order approximation of the model. Our resulting method SP2 uses Hessian-vector products to speed-up the convergence of SP. Furthermore, and rather uniquely among second-order methods, the design of SP2 in no way relies on positive definite Hessian matrices or convexity of the objective function. We show SP2 is very competitive on matrix completion, non-convex test problems and logistic regression. We also provide a convergence theory on sums-of-quadratics.

## 1 INTRODUCTION

Consider the problem

$$w^* \in \arg\min_{w \in \mathbb{R}^d} \left\{ f(w) := \frac{1}{n} \sum_{i=1}^{n} f_i(w) \right\}, \tag{1}$$

where $f$ is twice continuously differentiable, and the set of minimizers is nonempty. Let the optimal value of equation 1 be $f^* \in \mathbb{R}$, and $w^0$ be a given initial point. Here each $f_i(w)$ is the loss of a model parametrized in $w \in \mathbb{R}^d$ over an $i$-th data point. Our discussion, and forth coming results, also hold for a loss given as an expectation $f(w) = \mathbf{E}_{\xi \sim \mathcal{D}}[f_\xi(w)]$, where $\xi \sim \mathcal{D}$ is the data generating process and $f_\xi(w)$ the loss over this sampled data point. But for simplicity we use the $f_i(w)$ notation.

Contrary to classic statistical modeling, there is now a growing trend of using overparametrized models that are able to interpolate the data Ma et al. (2018); that is, models for which the loss is minimized over every data point as described in the following assumption.

**Assumption 1.** *We say that the interpolation condition holds when the loss is nonnegative, $f_i(w) \geq 0$,*

$$\text{and there exists} \quad w^* \in \mathbb{R}^d \quad \text{such that} \quad f(w^*) = 0. \tag{2}$$

*Consequently, $f_i(w^*) = 0$ for $i = 1, \ldots, n$.*

Overparameterized deep neural networks are the most notorious example of models that satisfy Assumption 1. Indeed, with sufficiently more parameters than data points, we are able to simultaneously minimize the loss over all data points.

If we admit that our model can interpolate the data, then we have that our optimization problem equation 1 is equivalent to solving the system of nonlinear equations

$$f_i(w) = 0, \quad \text{for } i = 1, \ldots, n. \tag{3}$$

Since we assume $f_i(w) \geq 0$ any solution to the above is a solution to our original problem.

Recently, it was shown in Berrada et al. (2020); Gower et al. (2021b) that the Stochastic Polyak step size (SP) method Loizou et al. (2020); Polyak (1987)

$$w^{t+1} = w^t - \frac{f_i(w^t)}{\|\nabla f_i(w^t)\|^2} \nabla f_i(w^t) \tag{4}$$

directly solves the interpolation equations. Indeed, at each iteration $\mathtt{SP}$ samples a single $i$-th equation from equation 3, then projects the current iterate $w^t$ onto the linearization of this constraint, that is

$$w^{t+1} = \arg\min_{w \in \mathbb{R}^d} \left\| w - w^t \right\|^2 \quad \text{s.t. } f_i(w^t) + \langle \nabla f_i(w^t), w - w^t \rangle = 0. \tag{5}$$

Here we take one step further, and instead of projecting onto the linearization of $f_i(w)$ we use the local quadratic expansion. That is, as a proxy of setting $f_i(w) = 0$ we set the quadratic expansion of $f_i(w)$ around $w^t$ to zero

$$f_i(w^t) + \langle \nabla f_i(w^t), w - w^t \rangle + \tfrac{1}{2} \langle \nabla^2 f_i(w^t)(w - w^t), w - w^t \rangle = 0. \tag{6}$$

The above quadratic constraint could have infinite solutions, a unique solution or no solution at all[1]. Indeed, for example if $\nabla^2 f_i(w^t)$ is positive definite, there may exist no solution, which occurs when $f_i$ is convex, and is the most studied setting for second order methods. But if the loss is positive $f_i$ and the Hessian has at least one negative eigenvalue, then equation 6 always has a solution.

If equation 6 has solutions, then analogously to the $\mathtt{SP}$ method, we can choose one using a projection step [2]

$$w^{t+1} \in \arg\min_{w \in \mathbb{R}^d} \tfrac{1}{2} \left\| w - w^t \right\|^2$$
$$\text{s.t. } f_i(w^t) + \langle \nabla f_i(w^t), w - w^t \rangle + \tfrac{1}{2} \langle \nabla^2 f_i(w^t)(w - w^t), w - w^t \rangle = 0. \tag{7}$$

We refer to equation 7 as the $\mathtt{SP2}$ method. Using a quadratic expansion has several advantages. First, quadratic expansions are more accurate than linearizations, which will allow us to take larger steps. Furthermore, using the quadratic expansion will lead to convergence rates which are *independent* on how well conditioned the Hessian matrices are, as we show later in Proposition 1.

Our $\mathtt{SP2}$ method occupies a unique position in the literature of stochastic second order method since it is incremental and in no way relies on convexity or positive semi-definite Hessian matrices. Indeed, as we will show in our non-convex experiments in 6.1 and matrix completition B, the $\mathtt{SP2}$ excels at minimizing non-convex problems that satisfy interpolation. In contrast, Newton based methods often converge to stationary points other than the global minima.

We also relax the interpolation assumption, and develop analogous quadratic methods for finding $w$ and the smallest possible $s \in \mathbb{R}$ such that

$$f_i(w) \leq s, \quad \text{for } i = 1, \ldots, n. \tag{8}$$

We refer to this as the *slack interpolation* equations, which were introduced in Crammer et al. (2006) for linear models. If the interpolation assumption holds then $s = 0$ and the above is equivalent to solving equation 3. When interpolation does not hold, then equation 8 is still a upper approximation of equation 1, as detailed in Gower et al. (2022).

The rest of this paper is organized as follows. We introduce some related work in Section 2. We present the proposed $\mathtt{SP2}$ methods in Section 3 and corresponding convergence analysis in Section 4. In Section 5, we relax the interpolation condition and develop a variety of quadratic methods to solve the slack version of this problem. We test the proposed methods with a series of experiments in Section 6. Finally, we conclude our work and discuss future directions in Section 7.

## 2 RELATED WORK

Since it became clear that Stochastic Gradient Descent (SGD), with appropriate step size tuning, was an efficient method for solving the training problem equation 1, there has been a search for an efficient second order counter part. The hope being, and our objective here, is to find a second order stochastic method that is *incremental*; that is, it can work with mini-batches, requires little to *no tuning* since it would depend less on how well scaled or conditioned the data is, and finally, would also apply to *non-convex* problems. To date there is a vast literature on stochastic second order methods, yet none that achieve all of the above.

The subsampled Newton methods such as (Roosta-Khorasani & Mahoney, 2019; Bollapragada et al., 2018; Liu & Roosta, 2021; Erdogdu & Montanari, 2015; Kohler & Lucchi, 2017; Jahani et al., 2017)

---

[1]Or even two solutions in the 1d case.

[2]Note that there could be more than one solution to this projection and we can choose one either with least norm or arbitrarily.

require large batch sizes in order to guarantee that the subsampled Newton direction is close to the full Newton direction in high probability. As such are not incremental. Other examples of large sampled based methods include the Stochastic quasi-Newton methods (Byrd et al., 2011; Mokhtari & Ribeiro, 2015; Moritz et al., 2016; Gower et al., 2016; Wang et al., 2017; Berahas et al., 2016), stochastic cubic Newton Tripuraneni et al. (2018), SDNA (Qu et al., 2016), Newton sketch (Pilanci & Wainwright, 2017) and Lissa (Agarwal et al., 2017), since these require a large mini-batch or full gradient evaluations.

The only incremental second order methods we aware of are *IQN* (Incremental Quasi-Newton) (Mokhtari et al., 2018), *SNM* (Stochastic Newton Method) (Kovalev et al., 2019; Rodomanov & Kropotov, 2016) and very recently *SAN* (Stochastic Average Newton) Chen et al. (2021). IQN and SNM enjoy a fast local convergence, but their computational and memory costs per iteration, is of $\mathcal{O}(d^2)$ making them prohibitive in large dimensions.

Handling non-convexity in second order methods is particularly challenging because most second order methods rely on convexity in their design. For instance, the classic Newton iteration is the minima of the local quadratic approximation if this approximation is convex. If it is not convex, the Newton step can be meaningless, or worse, a step uphill. Quasi-Newton methods maintain positive definite approximation of the Hessian matrix, and thus are also problematic when applied to non-convex problems Wang et al. (2017) for which the Hessian is typically indefinite. Furthermore the incremental Newton methods IQN, SNM and SAN methods rely on the convexity of $f_i$ in their design. Indeed, without convexity, the iterates of IQN, SNM and SAN are not well defined.

In contrast, our approach of finding roots of the local quadratic approximation equation 7 in no way relies on convexity, and relies solely on the fact that the local quadratic approximation around $w^t$ is good if we are not far from $w^t$. But our approach does introduce a new problem: the need to solve a system of quadratic equations. We propose a series of methods to solve this in Sections 3 and 5.

Solving quadratic equations has been heavily studied. There are even dedicated methods for solving

$$w^\star = \operatorname*{argmin}_{w \in \mathbb{R}^d} \tfrac{1}{2}\|w - \bar{w}\|^2 \ \text{ s.t. } Q(w) = 0, \ \text{ where } Q(w) = \tfrac{1}{2}w^\top \mathbf{H}w + b^\top w + c \qquad (9)$$

for a given $\bar{w}$, where $\mathbf{H}$ is a nonzero symmetric (*not necessarily PSD*) matrix, and the level set $\{w : Q(w) = 0\}$ is nonempty. Note that since $Q(w)$ is a quadratic function, the problem equation 9 is nonconvex. Yet despite this non-convexity, so long as there exists a feasible point, the projection equation 9 can be solved in polynomial time by re-writing the projection as a semi-definite program, or by using the S-procedure, which involves computing the eigenvalue decomposition of $\mathbf{H}$ and using a line search as proposed in Park & Boyd (2017), and detailed here in Section A. But this approach is too costly when the dimension $d$ is large.

An alternative iterative method is proposed in Sosa & MP Raupp (2020), but only asymptotic convergence is guaranteed. In Dai (2006), the authors consider a similar problem by projecting a point onto a general ellipsoid, which is again a problem of solving quadratic equations. However, they require the matrix $\mathbf{H}$ to be a positive definite matrix.

The problem equation 7 and equation 9 are also an instance of a quadratic constrained quadratic program (QCQP). Although the QCQP in equation 7 has no closed form solution in general, we show in the next section that there is a closed form solution for Generalized linear models (GLMs), that holds for convex and non-convex GLMs alike. For general non-linear models we propose in Section 3.2 an approximate solution to equation 7 by iteratively linearizing the quadratic constraint and projecting onto the linearization.

## 3 THE SP2 METHOD

Next we give a closed form solution to equation 7 for GLMs. We then provide an approximate solution to equation 7 for more general models.

### 3.1 SP2 - GENERALIZED LINEAR MODELS

Consider when $f_i$ is the loss over a linear model with

$$f_i(w) = \phi_i(x_i^\top w - y_i), \qquad (10)$$

where $\phi_i : \mathbb{R} \to \mathbb{R}$ is a loss function, and $(x_i, y_i) \in \mathbb{R}^{d+1}$ is an input-output pair. Consequently

$$\nabla f_i(w) = \phi_i'(x_i^\top w - y_i)x_i := a_i x_i, \qquad \nabla^2 f_i(w) = \phi_i''(x_i^\top w - y_i)x_i x_i^\top := h_i x_i x_i^\top. \qquad (11)$$

The quadratic constraint problem equation 7 can be solved exactly for GLMs equation 10 as we show next.

**Lemma 1.** *(SP2) Assume $f_i(w)$ is the loss of a generalized linear model equation 10 and is non-negative. Let $f_i = f_i(w^t)$ for short. Let $a_i := \phi_i'(x_i^\top w - y_i)$ and $h_i := \phi_i''(x_i^\top w - y_i)$. If*

$$a_i^2 - 2h_i f_i \geq 0 \tag{12}$$

*then the optimal solution of equation 7 is as follows*

$$w^{t+1} = w^t - \frac{a_i}{h_i}\left(1 - \frac{\sqrt{a_i^2 - 2h_i f_i}}{|a_i|}\right)\frac{x_i}{\|x_i\|^2}. \tag{13}$$

*Alternatively if equation 12 does not hold, since $f_i \geq 0$ we have necessarily that $h_i > 0$, and consequently a Newton step will give the minima of the local quadratic, that is*

$$w^{t+1} = w^t - \frac{a_i}{h_i}\frac{x_i}{\|x_i\|^2}. \tag{14}$$

The proof to the above lemma, and all subsequent missing proofs can be found in the appendix. Lemma 1 establishes a condition equation 12 under which we should not take a full Newton step. Interestingly, this condition equation 12 holds when the square root of the loss function has negative curvature, as we show in the next lemma.

**Lemma 2.** *Let $\phi$ be a non-negative function which is twice differentiable at all $t$ with $\phi(t) \neq 0$. The condition equation 12, in other words*

$$\phi'(t)^2 \geq 2\phi(t)\phi''(t)$$

*holds when $\sqrt{\phi(t)}$ is concave away from its roots, i.e. when $\frac{d^2}{dt^2}\sqrt{\phi(t)} \leq 0$ for all $t$ with $\phi(t) \neq 0$.*

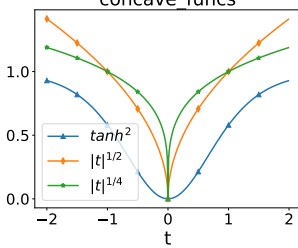

Examples of loss functions include $\phi(t) = \tanh^2(t)$ and $\phi(t) = t^p$ with $0 \leq p \leq 2$ (see Figure **(a)**).

**Figure (a)** Non-convex loss func. for which condition equation 12 holds.

In conclusion to this section, SP2 has a closed form solution for GLMs, and this closed form solution includes many non-convex loss functions.

## 3.2 SP2$^+$ - Linearizing and Projecting

In general, there is no closed form solution to equation 7. Indeed, there may not even exist a solution. Inspired by the fact that computing a Hessian-vector product can be done with a single backpropagation[3] at the same cost as computing a gradient Christianson (1992), we will make use of the cheap Hessian-vector product to derive an approximate solution to equation 7.

Instead of solving equation 7 exactly, here we propose to take two steps towards solving equation 7 by projecting onto the linearized constraints. To describe this method let

$$q(w) := f_i(w^t) + \langle \nabla f_i(w^t), w - w^t \rangle + \frac{1}{2}\langle \nabla^2 f_i(w^t)(w - w^t), w - w^t \rangle. \tag{15}$$

In the first step we linearize the quadratic constraint equation 15 around $w^t$ and project onto this linearization:

$$w^{t+1/2} = \arg\min_{w \in \mathbb{R}^d} \frac{1}{2}\|w - w^t\|^2 \quad \text{s.t. } f_i(w^t) + \langle \nabla f_i(w^t), w - w^t \rangle = 0. \tag{16}$$

The closed form update of this first step is given by

$$w^{t+1/2} = w^t - \frac{f_i(w^t)}{\|\nabla f_i(w^t)\|^2}\nabla f_i(w^t), \tag{17}$$

which is a Stochastic Polyak step equation 4. For the second step, we once again linearize the quadratic constraint equation 15, but this time around the point $w^{t+1/2}$ and set this linearization to zero, that is

$$w^{t+1} = \arg\min_{w \in \mathbb{R}^d} \frac{1}{2}\|w - w^{t+1/2}\|^2 \quad \text{s.t. } q(w^{t+1/2}) + \langle \nabla q(w^{t+1/2}), w - w^{t+1/2} \rangle = 0. \tag{18}$$

---

[3]We use `torch.autograd.functional.hvp` in our code to compute this Hessian-vector product.

The closed form update of this second step is given by

$$w^{t+1} = w^{t+1/2} - \frac{q(w^{t+1/2})}{\left\|\nabla q(w^{t+1/2}))\right\|^2} \nabla q(w^{t+1/2}). \tag{19}$$

We refer to the resulting proposed method as the SP2$^+$ method, summarized in the following.

**Lemma 3.** *(SP2$^+$) Let $g_t \equiv \nabla f_i(w^t)$ and $\mathbf{H}_t \equiv \nabla^2 f_i(w^t)$. Update equation 17 and equation 19 is given by*

$$w^{t+1} = w^t - \frac{f_i(w^t)}{\|g_t\|^2} g_t - \frac{1}{2} \frac{f_i(w^t)^2}{\|g_t\|^4} \frac{\langle \mathbf{H}_t g_t, g_t \rangle}{\|v^{t+1}\|^2} v^{t+1}, \text{ where } v^{t+1} = \left(\mathbf{I} - \mathbf{H}_t \frac{f_i(w^t)}{\|g_t\|^2}\right) g_t, \tag{20}$$

In equation 20 we can see that SP2$^+$ applies a second order correction to the SP step.

SP2$^+$ is equivalent to two steps of a Newton Raphson method applied to finding a root of $q(w)$. If we apply multiple steps of the Newton Raphson method, as opposed to two, the resulting method converges to the root of $q$, see Theorem 2 in the appendix. Theorem 2 shows that this multi-step version of SP2$^+$ converges when $q$ belongs to a large class of non-convex functions known as the star-convex functions. Star-convexity, which is a generalization of convexity, includes several non-convex loss functions Hinder et al. (2019).

## 4 CONVERGENCE THEORY

Here we provide a convergence theory for SP2 and SP2$^+$ for when $f(w)$ is an average of quadratic functions. Fix $w^* \in \mathbb{R}^d$ and let the loss over the $i$-th data point be given by

$$f_i(w) = \langle \mathbf{H}_i(w - w^*), w - w^* \rangle, \tag{21}$$

where $\mathbf{H}_i \in \mathbb{R}^{d \times d}$ is a symmetric positive semi-definite matrix for $i = 1, \ldots, n$. Note that we assume that our algorithms have access to the second order terms $\mathbf{H}_i$, but not to the minimizer $w^*$. Consequently $f_i(w^*) = 0 = f(w^*) = \min_{w \in \mathbb{R}^d} f(w)$, thus the interpolation condition holds.

**Proposition 1.** *Consider the loss functions given in equation 21. The SP2 method equation 7 converges linearly*[4]

$$\mathbb{E}\|w^{t+1} - w^*\|^2 \le \rho \, \mathbb{E}\|w^t - w^*\|^2, \quad \text{where } \rho = \lambda_{\max}(\mathbf{I} - \frac{1}{n}\sum_{i=1}^{n} \mathbf{H}_i \mathbf{H}_i^+) < 1. \tag{22}$$

The rate of convergence of SP2 in equation 22 can be orders of magnitude better than SGD. Indeed, since equation 21 is convex, smooth and interpolation holds, we have from Needell et al. (2016) that SGD converges at a rate of

$$\rho_{SGD} = 1 - \frac{1}{2n} \frac{\lambda_{\min}(\sum_{i=1}^{n} \mathbf{H}_i)}{\max_{i=1,\ldots,n} \lambda_{\max}(\mathbf{H}_i)}. \tag{23}$$

To compare equation 23 to $\rho$ rate in Proposition 1, consider the case where all $\mathbf{H}_i$ are invertible. In this case $\mathbf{H}_i \mathbf{H}_i^+ = \mathbf{I}$ and thus $\rho = 0$ and SP2 converges in one step. Indeed, even if a single $\mathbf{H}_i$ is invertible, after sampling $i$ the SP2 method will reach $w^*$, solving the system exactly. Of course the method is more interesting when the $\mathbf{H}_i$'s are low rank . In contrast, the SGD method is still at the mercy of the spectra of the $\mathbf{H}_i$ matrices and depend on how well conditioned these matrices are. Even in the extreme case where all $\mathbf{H}_i$ are well conditioned, for example $\mathbf{H}_i = i \times \mathbf{I}$, the rate of convergence of SGD can be very slow, for instance in this case we have $\rho_{SGD} = 1 - \frac{1}{2n^2}$.

**Proposition 2.** *Consider the loss functions in equation 21. The SP2$^+$ method equation 20 converges linearly*

$$\mathbb{E}\|w^{t+1} - w^*\|^2 \le \rho_{SP2+}^2 \, \mathbb{E}\|w^t - w^*\|^2, \text{ where } \rho_{SP2+} = 1 - \frac{1}{2n}\sum_{i=1}^{n} \frac{\lambda_{\min}(\mathbf{H}_i)}{\lambda_{\max}(\mathbf{H}_i)}. \tag{24}$$

The proof of Proposition 2 follows from Corollary 5.7 in Gower et al. (2021a). The rate of convergence of SP2$^+$ now depends on the average condition number of the $\mathbf{H}_i$ matrix. The rate in equation 24 can be greater or smaller than the rate of SGD in equation 23. For instance, if one $\mathbf{H}_j$ has a condition number that is much greater, then equation 24 will be smaller than equation 23. On

---

[4]Thanks to an anonymous reviewer, the proof technique we use for Proposition 1 is the same proof technique used for the Block Kaczmarz method Needell & Tropp (2014), see Remark 1.

the other hand, if all the $\mathbf{H}_j$'s are such that $\max_{i=1,\ldots,n} \lambda_{\max}(\mathbf{H}_i) = \lambda_{\max}(\mathbf{H}_j)$, for $j = 1,\ldots,n$, then equation 23 is a smaller rate as compared to equation 24.

Note also that the rate of $\mathtt{SP2}^+$ appears squared in equation 24 and the rate $\rho_{SGD}$ of SGD is not squared. But this difference accounts for the fact that each step of $\mathtt{SP2}^+$ is at least twice the cost of SGD, since each step of $\mathtt{SP2}^+$ is comprised of two gradient steps, see equation 17 and equation 19. Thus we can neglect the apparent advantage of the rate $\rho_{SP2+}$ being squared.

## 5  QUADRATIC WITH SLACK

Here we depart from the interpolation Assumption 1 and design a variant of $\mathtt{SP2}^+$ that can be applied to models that are *close* to interpolation. Instead of trying to set all the losses to zero, we now will try to find the smallest *slack variable* $s > 0$ for which

$$f_i(w) \le s, \quad \text{for } i = 1,\ldots,n.$$

If interpolation holds, then $s = 0$ is a solution. Outside of interpolation, $s$ may be non-zero. There are two natural ways of translating the slack constraint above into a penalty term. Namely one can consider either the associated $\ell_2$ or $\ell_1$ loss. We explore both of these in turn.

### 5.1  L2 SLACK FORMULATION

To make $s$ as small as possible, we consider the following problem

$$\min_{s \in \mathbb{R}, w \in \mathbb{R}^s} \frac{1}{2} s^2 \text{ subject to } f_i(w) \le s, \text{ for } i = 1,\ldots,n, \tag{25}$$

which is called the *L2 slack formulation*. This type of slack problem was introduced in Crammer et al. (2006) to derive variants of the passive-aggressive method that apply to linear models on non-separable data; in other words, when the models cannot interpolate the data.

To solve equation 25 we will again project onto a local quadratic approximation of the constraint. Let

$$q_{i,t}(w) := f_i(w^t) + \left\langle \nabla f_i(w^t), w - w^t \right\rangle + \tfrac{1}{2} \left\langle \nabla^2 f_i(w^t)(w - w^t), w - w^t \right\rangle. \tag{26}$$

and let $\Delta_t = \|w - w^t\|^2 + (s - s^t)^2$. Consider the iterative method given by

$$w^{t+1}, s^{t+1} = \underset{s \ge 0,\ w \in \mathbb{R}^d}{\arg\min} \frac{1-\lambda}{2} \Delta_t + \frac{\lambda}{2} s^2 \quad \text{s.t. } q_{i,t}(w) \le s, \tag{27}$$

where $\lambda \in [0,\ 1]$ is a regularization parameter that trades off between having a small $s$, and using the previous iterates as a regularizer. The resulting projection problem in equation 27 has a quadratic inequality, and thus in most cases has no closed form solution, despite always being feasible[5].

So instead of solving equation 27 exactly, we propose an approximate solution by iteratively linearizing and projecting onto the constraints. Our approximate solution has two steps, the first step being

$$w^{t+1/2}, s^{t+1/2} = \underset{s \ge 0,\ w \in \mathbb{R}^d}{\arg\min} \frac{1-\lambda}{2} \Delta_t + \frac{\lambda}{2} s^2 \quad \text{s.t. } q_{i,t}(w^t) + \left\langle \nabla q_{i,t}(w^t), w - w^t \right\rangle \le s. \tag{28}$$

The second step is given by projecting $w^{t+1/2}$ onto the linearization around $w^{t+1/2}$ as follows

$$w^{t+1}, s^{t+1} = \underset{s \ge 0,\ w \in \mathbb{R}^d}{\arg\min} \frac{1-\lambda}{2} \Delta_{t+\frac{1}{2}} + \frac{\lambda}{2} s^2 \quad \text{s.t. } q_{i,t}(w^{t+1/2}) + \left\langle \nabla q_{i,t}(w^{t+1/2}), w - w^{t+1/2} \right\rangle \le s. \tag{29}$$

The closed form solution to our two step method is given in Lemma 7 of Appendix C.7. We refer to this method as $\mathtt{SP2L2}^+$.

### 5.2  L1 SLACK FORMULATION

To make $s$ as small as possible, we can also solve the following *L1 slack formulation*

$$\min_{s \ge 0, w \in \mathbb{R}^d} s \quad \text{s.t. } f_i(w) \le s, \text{ for } i = 1,\ldots,n.$$

We can again project onto a local quadratic approximation of the constraint. That is, let $\lambda \in [0,\ 1]$ be a regularization parameter that trades off between having a small $s$, and using the previous iterates as

---

[5]For instance $w = w^t$ and $s = f_i(w^t)$ is feasible

a regularizer, and consider the iterative method given by

$$w^{t+1}, s^{t+1} = \underset{s \geq 0, \, w \in \mathbb{R}^d}{\arg\min} \frac{1-\lambda}{2} \Delta_t + \frac{\lambda}{2} s \quad \text{s.t. } q_{i,t}(w) \leq s, \tag{30}$$

To approximately solve equation 30, we again propose an approximate two step method similar to equation 28 and equation 29. The closed form solution to the two step method is given in Lemma 9 of Appendix C.8. We refer to this method as $\texttt{SP2L1}^+$.

### 5.3 Dropping the Slack Regularization

Note that the objective function in equation 30 contains a regularization term $(s - s^t)^2$, which forces $s$ to be close to $s^t$. If we allow $s$ to be far from $s^t$, we can instead solve the following unregularized problem

$$w^{t+1}, s^{t+1} = \underset{s \geq 0, \, w \in \mathbb{R}^d}{\arg\min} \frac{1-\lambda}{2} \left\| w - w^t \right\|^2 + \frac{\lambda}{2} s \quad \text{s.t. } q_{i,t}(w) \leq s, \tag{31}$$

where $\lambda \in [0, \, 1]$ is again a regularization parameter that trades off between having a small $s$, and using the previous iterates as a regularizer. We call the resulting method in equation 31 the $\texttt{SP2max}$ method since it is a second order variant of the $\texttt{SPmax}$ method Loizou et al. (2020); Gower et al. (2022). The advantage of $\texttt{SP2max}$ is that it has a closed form solution for GLMs in the form of equation 10. We give the closed form in the following lemma which is proved in Appendix C.10.

**Lemma 4.** *($\texttt{SP2max}$) Consider the GLM model given in equation 10 and equation 11. If the loss $f_i = f_i(w^t)$ is non-negative, then the iterates of equation 31 have a closed form solution given by*

$$w^{t+1} = w^t + c^\star x_i, \qquad s^{t+1} = \max\{\widetilde{s}, \, 0\}, \quad c^\star = \begin{cases} 0, & \text{if } f_i = 0 \\ -\dfrac{\widetilde{\lambda} a_i}{1 + \widetilde{\lambda} h_i \ell}, & \text{if } f_i > 0 \text{ and } \widetilde{s} \geq 0, \\ \dfrac{-a_i + \sqrt{a_i^2 - 2 h_i f_i}}{h_i \ell}, & \text{otherwise.} \end{cases}$$

*and where $\widetilde{s} = f_i - \dfrac{\widetilde{\lambda} a_i^2 \ell}{1 + \widetilde{\lambda} h_i \ell} + \dfrac{h_i \widetilde{\lambda}^2 a_i^2 \ell^2}{2(1 + \widetilde{\lambda} h_i \ell)^2}$, and $\ell = \|x_i\|^2$, $\widetilde{\lambda} = \dfrac{\lambda}{2(1-\lambda)}$.*

To approximately solve equation 31 in general, we again propose an approximate two step method. The closed form solution to the two step method is given in Lemma 10 of Appendix C.9. We refer to this method as $\texttt{SP2max}^+$.

## 6 Experiments

### 6.1 Non-convex problems

To emphasize how our new $\texttt{SP2}$ methods can handle non-convexity, we have tested $\texttt{SP2}$ equation 7 (blue curve) , $\texttt{SP2}^+$ equation 20 (green curve) on the non-convex problems PermD$\beta^+$, Rastrigin and Levy N. 13, Rosenbrock Jamil & Yang (2013)[6], see Figures 1, 2 in the main text, and Figures 6 and 7 in the appendix. For a baseline we compared against $\texttt{SGD}$ (yellow ), Adam (pink) and $\texttt{Newton}$ (red).

The two experiments with the function Levy N. 13 and Rosenbrock are detailed in the appendix in Section D.1.2.

All of these functions are 2D sums-of-terms of the format equation 1 and satisfy the interpolation Assumption 1. To compute the $\texttt{SP2}$ update we used ten steps of Newton's Raphson method as detailed in Section C.4. We consistently find across these non-convex problems that $\texttt{SP2}$ and $\texttt{SP2}^+$ are very competitive, with $\texttt{SP2}$ converging in under 20 epochs. $\texttt{SP2}$ also converges faster in terms of time taken (see middle of Figures 1 and 2). Here we can clearly see that $\texttt{SP2}$ converges to a high precision solution (like most second order methods), and different than other second order methods is not attracted to local maxima or saddle points. In contrast, $\texttt{Newtons}$ method converges to a local maxima on all problems excluding the Rosenbrock function in Figure 7 in the appendix. For instance on the right of Figure 2 we can see the red dot of $\texttt{Newton}$ stuck on a local maxima.

---

[6]We used the Python Package `pybenchfunction` available on github `Python_Benchmark_Test_ Optimization_Function_Single_Objective`. We also note that the PermD$\beta^+$ implemented in this package is a modified version of the `PermD`$\beta$ function, as we detail in Section D.1.1.

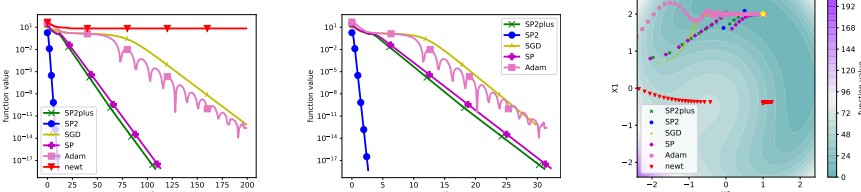

Figure 1: The 2D PermD$\beta^+$ function with $\beta = 0.5$ we plot Left: $f(x)$ vs epochs, Middle: $f(x)$ vs time, Right: level set plot, Right: Surface plot.

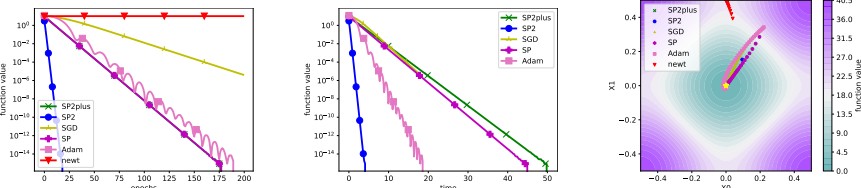

Figure 2: The 2D Rastrigin function where Left: we plot $f(x)$ across epochs Middle: level set plot, Right: Surface plot.

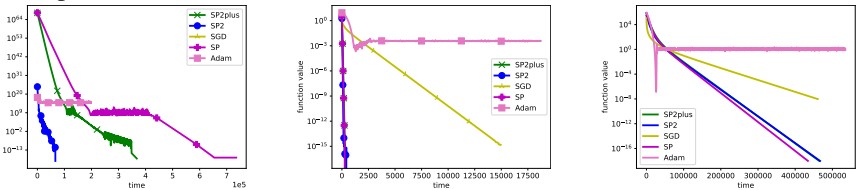

Figure 3: Function value vs time (seconds) for the PermD$\beta^+$ ($d = 10$) function with $\beta = 0.5$ (Left), the Rastrigin function ($d = 100$) (Middle) and the Rosenbrock function ($d = 100$).

When increasing the dimension to 100, these non-convex problems become very challenging, see Figure 3[7], and often only SP2, SP2$^+$ and SP are able to find the global minima. Note that we no longer compare to Newton's method because it was exceeding our maximum allocated time.

## 6.2 MATRIX COMPLETION

Assume a set of known values $\{a_{i,j}\}_{(i,j)\in\Omega}$ where $\Omega$ is a set of known elements of the matrix, and we want to determine the missing elements. One approach is solving the *matrix completion* problem

$$\min_{U,V} \sum_{(i,j)\in\Omega} \tfrac{1}{2}(u_i^T v_j - a_{i,j})^2, \tag{32}$$

where $A = [a_{i,j}] \in \mathbb{R}^{m\times n}$, $U = [u_i]_{i=1,\ldots,m} \in \mathbb{R}^{r\times m}$ and $V = [v_j]_{j=1,\ldots,n} \in \mathbb{R}^{r\times n}$. After solving equation 32, we then use the matrix $U^\top V$ as an approximation to the complete matrix $A$. Despite equation 32 being a non-convex problem, if there exists an *interpolating* solution to equation 32, one where $u_i^T v_j = a_{i,j}$, for $(i,j) \in \Omega$, then the SP2 method can solve equation 32. Indeed, SP2 can be applied to equation 32 by sampling a single pair $(i,j) \in \Omega$ uniformly, then projecting onto the quadratic (the solution to which is detailed in Theorem 1 in Appendix B):

$$u_i^{k+1}, v_j^{k+1} = \operatorname*{arg\,min}_{u,v} \tfrac{1}{2}\left\|u - u_i^k\right\|^2 + \tfrac{1}{2}\left\|v - v_j^k\right\|^2 \text{ subject to } u^\top v = a_{i,j}. \tag{33}$$

We compared our method 33 to a specialized variant of SGD for online matrix completion described in Jin et al. (2016), see Figure 4. To compare the two methods we generated a rank $r = 2$ matrix $A \in \mathbb{R}^{100\times 50}$. We selected a subset entries with probability $p = 0.1, 0.2$ or $0.3$ to form our set $\Omega_{init}$ that was used to obtain an initial estimate $U_0, V_0$ using rank-k SVD method as described in Jin et al. (2016). We extensively tuned the step size of SGD using a grid search, and the method labelled Non-convex SGD is the resulting run of SGD with the best step size. We also show how sensitive SGD is to this step size, by including the run of SGD with step sizes that were only a factor of 2 to 4 away from the optimal, which greatly degrades the performance of SGD. In contrast, SP2 worked with no tuning, and matches the performance of SGD with the optimal step size in the $p = 0.1$ experiment, and outperforms SGD when more measurements are available in the $p = 0.2$ and $p = 0.3$ figures.

---

[7]See Figure 9 for a comparison with respect to epochs.

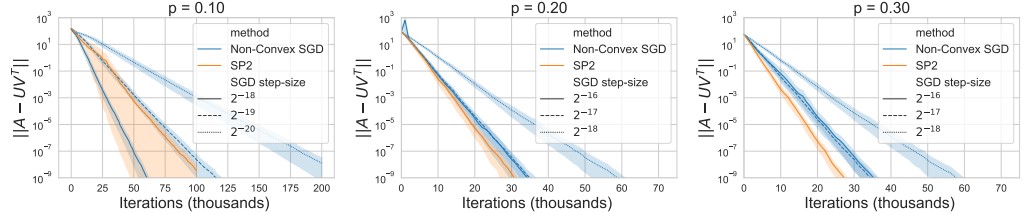

Figure 4: Recovery error for matrix completion. Left: Using 10%, Middle: using 20% and Right: using 30% of the entries of $A$ to form $\Omega$, respectively. Shaded region corresponds to 5 repeated runs.

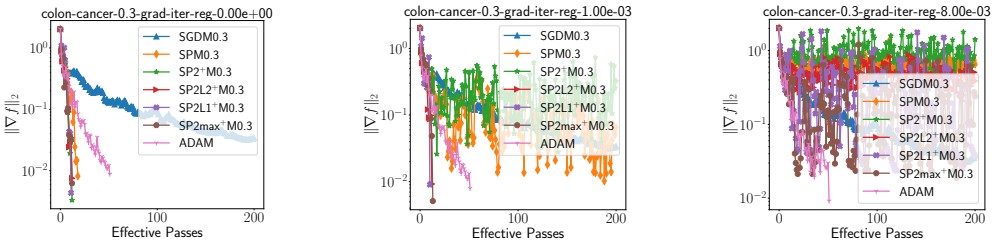

Figure 5: Colon-cancer: gradient norm at each epoch with momentum set at 0.3. Left: $\sigma = 0$, Middle: $\sigma = 0.001$ and Right $\sigma = 0.008$.

### 6.3 CONVEX CLASSIFICATION

Here we compare our proposed methods on a logistic regression problem to SGD, SP, and ADAM. In particular, we consider the problem of minimizing the following loss function $f(w) = \frac{1}{n} \sum_{i=1}^{n} f_i(w) + \frac{\sigma}{2} \|w\|_2^2$, where $f_i(w) = \phi_i(x_i^\top w)$ with $\phi_i(t) = \ln(1 + e^{-y_i t})$. Here, $\{(x_i, y_i) \in \mathbb{R}^{d+1}\}_{i=1}^{n}$ stands for the feature-label pairs and $\sigma > 0$ is the regularization parameter. We control how far each problem is from interpolation by increasing $\sigma$. When $\sigma > 0$ the problem cannot interpolate, and thus we expect to see a benefit of the slack methods in Section 5 over SP2$^+$. We used two data sets: colon-cancer ($(n, d) = (62, 2000)$) Alon et al. (1999) and mushrooms ($(n, d) = (8124, 112)$) West et al. (2001), both of which interpolate when $\sigma = 0$.

We compare the proposed methods SP2$^+$ equation 20, SP2L2$^+$ (Lemma 7), SP2L1$^+$ (Lemma 9), and SP2max$^+$ (Lemma 10) with SGD, SP equation 4, and ADAM on both data sets with three regularizations $\sigma \in \{0, 0.001, 0.008\}$ and with momentum set to 0.3. For the SGD method, we use a learning rate $L_{\max}/\sqrt{t}$ in the $t$-th iteration, where $L_{\max} = \frac{1}{4} \max_{i=1,\dots,n} \|x_i\|^2$ denotes the smoothness constant of the loss function. We chose $\lambda$ for SP2L2$^+$, SP2L1$^+$, and SP2max$^+$ using a grid search of $\lambda \in \{0.1, 0.2, \dots, 0.9\}$, the details are in Section D.

The gradient norm evaluated at each epoch is presented in Figures 5 and 16 (see Appendix D). We see that SP2 methods converge much faster than classical methods (e.g., SGD, SP, ADAM) and need fewer epochs to achieve the tolerance when $\sigma$ is small (left and middle plots). However, they can all fail when the problem is far from interpolation, e.g., when $\sigma = 8 \times 10^{-3}$. The running time used for each algorithm to achieve the tolerance for both data sets is presented in Figure 17 (see Appendix D).

## 7 CONCLUSION

We have proposed new second order methods aimed at overparameterized models that can interpolate (or nearly interpolate) the data. In contrast to previous incremental second order methods, ours do not rely on convexity. Quite the opposite, the SP2 method can benefit from the Hessian having at least one negative eigenvalue. Consequently the SP2 method excels at optimizing non-convex models that interpolate, as we demonstrated in Sections 6.1 and B. We also provided a convergence result in Theorem 2 showing that SP2 can converge significantly faster than SGD for sums-of-quadratics. In Section 5, we then developed second order methods that solve a relaxed version of the interpolation equations by allowing some slack. We showed that these methods still perform well on problems that are close to interpolation in Section 6.3. In future work, it would be interesting to develop specialized variants of SP2 for optimizing Deep Neural Networks (DNNs). DNNs are particularly well suited since modern deep models often interpolate, are non-convex, and enjoy fast gradient and Hessian-vector computations via back-propagation.

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

## A   PROJECTING ONTO QUADRATIC

This following projection lemma is based on Section B in Park & Boyd (2017). What we do in addition to Park & Boyd (2017) is to clarify how to compute the resulting projection, and add further details on the proof.

**Lemma 5.** *Let $w \in \mathbb{R}^d$ and let $\mathbf{P} \in \mathbb{R}^{d \times d}$ is a symmetric matrix. Consider the projection*

$$w' \in \underset{w \in \mathbb{R}^d}{\arg\min} \ \tfrac{1}{2} \left\| w - z \right\|^2$$
$$s.t. \ r + \langle q, w - z \rangle + \tfrac{1}{2} \langle \mathbf{P}(w-z), w-z \rangle = 0. \tag{34}$$

*Let*

$$\lambda_1 \leq \lambda_2 \leq \cdots \leq \lambda_d$$

*be the eigenvalues of $\mathbf{P}$ and let $Q \Lambda Q^\top = \mathbf{P}$ be the eigenvalue decomposition of $\mathbf{P}$, where $\Lambda = diag(\lambda_i)$ and $QQ^\top = I$. Let $\hat{q} = Q^\top q$. If the quadratic constraint in equation 34 is feasible, then there exists a solution to equation 34. Now we give the three candidate solutions.*

*1. If $r = 0$, then the solution is given by*

$$w = z.$$

*2. Now assuming $r \neq 0$. Let*

$$\nu \quad = \quad \max_{i \, : \, \lambda_i \neq 0} \left\{ -\frac{1}{\lambda_1}, -\frac{1}{\lambda_d} \right\} \tag{35}$$

$$i^* \quad \in \quad \arg \max_{i \, : \, \lambda_i \neq 0} \left\{ -\frac{1}{\lambda_1}, -\frac{1}{\lambda_d} \right\} \tag{36}$$

$$N \quad = \quad \{ i \, : \, \lambda_i \neq \lambda_{i^*} \}. \tag{37}$$

*Let*

$$x^* \quad = \quad -(\mathbf{I} + \nu \Lambda)^\dagger \nu \hat{q}. \tag{38}$$

*If*

$$2\nu r + \nu \langle \hat{q}, x^* \rangle - \left\| x^* \right\|^2 + \frac{\nu^2}{4} \sum_{i \in N} \hat{q}_i^2 \geq 0 \tag{39}$$

*then the solution is given by*

$$w' = z + Q(x^* + n), \tag{40}$$

*where $n \in \mathbb{R}^d$ and*

$$n_i = \frac{\nu}{2} \hat{q}_i + \frac{1}{\sqrt{|N|}} \sqrt{2\nu r + \nu \langle \hat{q}, x^* \rangle - \left\| x^* \right\|^2 + \frac{\nu^2}{4} \sum_{i \in N} \hat{q}_i^2}, \quad \text{for } i \in N$$
$$n_i = 0, \quad \text{for } i \notin N.$$

3. *Alternatively if equation 39 does not hold, then the solution is given by*

$$w' \;=\; z - (\mathbf{I} + \nu\Lambda)^{\dagger}\nu q \tag{41}$$

*where $\nu$ is the solution to the nonlinear equation*

$$\frac{\nu}{2}\sum_i \frac{\hat{q}_i^2(2 + \nu\lambda_i)}{(1 + \nu\lambda_i)^2} \;=\; r. \tag{42}$$

*Proof.* First note that there exists a solution to equation 34 since the constraint is a closed feasible set.

Let $Q\Lambda Q^{\top} = \mathbf{P}$ be the SVD of $\mathbf{P}$, where $QQ^{\top} = I$. By changing variables $x = Q^{\top}(w - z)$ we have that equation 34 is equivalent to

$$\underset{\hat{x}\in\mathbb{R}^d}{\arg\min}\;\tfrac{1}{2}\,\|x\|^2$$
$$\text{s.t. } r + \langle \hat{q}, x\rangle + \tfrac{1}{2}\langle\Lambda x, x\rangle = 0, \tag{43}$$

where $\hat{x} = Q^{\top}(w - z)$ and $\hat{q} = Q^{\top}q$. The Lagrangian of equation 43 is given by

$$\begin{aligned}
L(x, \nu) &= \tfrac{1}{2}\,\|x\|^2 + \nu\big(r + \langle\hat{q}, x\rangle + \tfrac{1}{2}\langle\Lambda x, x\rangle\big) \\
&= \tfrac{1}{2}x^{\top}(\mathbf{I} + \nu\Lambda)x + \nu\big(r + \langle\hat{q}, x\rangle\big).
\end{aligned} \tag{44}$$

Thus the KKT conditions are given by

$$\nabla_x L(x, \nu) = (\mathbf{I} + \nu\Lambda)x + \nu\hat{q} = 0 \tag{45}$$
$$\nabla_\nu L(x, \nu) = r + \langle\hat{q}, x\rangle + \tfrac{1}{2}\langle\Lambda x, x\rangle = 0. \tag{46}$$

Since we are guaranteed that the projection has a solution, we have that as a necessary condition that the solution satisfies

$$\nabla_x^2 L(x, \nu) = (\mathbf{I} + \nu\Lambda) \succeq 0,$$

see Theorem 12.5 in Wright & Nocedal (1999). Consequently either $(\mathbf{I} + \nu\Lambda) \succ 0$ or $(\mathbf{I} + \nu\Lambda)$ has a zero eigenvalue.

Consider the case where $(\mathbf{I} + \nu\Lambda) \succ 0$. From equation 45 we have that

$$x = -\nu(\mathbf{I} + \nu\Lambda)^{-1}\hat{q}. \tag{47}$$

Now note that if $\nu = 0$ then $x = 0$ and by the constraint we must have $r = 0$. Otherwise, if $r \neq 0$, then $\nu \neq 0$. Assume now $\nu \neq 0$ and substituting the above into equation 46 and letting $\Lambda = \text{diag}(\lambda_i)$ gives

$$\begin{aligned}
\nabla_\nu L(x, \nu) &= r + \langle\hat{q}, x\rangle + \tfrac{1}{2\nu}\langle\nu\Lambda x, x\rangle \\
&= r + \langle\hat{q}, x\rangle + \tfrac{1}{2\nu}\langle(\mathbf{I} + \nu\Lambda)x, x\rangle - \frac{1}{2\nu}\|x\|^2 \\
&= r + \frac{1}{2}\langle\hat{q}, x\rangle - \frac{1}{2\nu}\|x\|^2 &&\text{Using equation 45} \\
&= r - \frac{\nu}{2}\langle\hat{q}, (\mathbf{I} + \nu\Lambda)^{-1}\hat{q}\rangle - \frac{\nu}{2}\|(\mathbf{I} + \nu\Lambda)^{-1}\hat{q}\|^2 &&\text{Using equation 47} \\
&= r - \frac{\nu}{2}\sum_i\left(\frac{\hat{q}_i^2}{1 + \nu\lambda_i} + \frac{\hat{q}_i^2}{(1 + \nu\lambda_i)^2}\right).
\end{aligned}$$

Thus

$$\frac{\nu}{2}\sum_i \frac{\hat{q}_i^2(2 + \nu\lambda_i)}{(1 + \nu\lambda_i)^2} \;=\; r. \tag{48}$$

Upon finding the solution $\nu$ to the above, we have that our final solution is given by $w' = z + Qx$, that is

$$\begin{aligned}
w' &= z - Q(\mathbf{I} + \nu\Lambda)^{\dagger}\nu\hat{q} \\
&= z - (\mathbf{I} + \nu\Lambda)^{\dagger}\nu Q\hat{q} \\
&= z - \nu(\mathbf{I} + \nu\Lambda)^{\dagger}q
\end{aligned} \tag{49}$$

Alternatively, suppose that $(\mathbf{I} + \nu\Lambda) \succeq 0$ is non-singular. The positive definiteness implies that

$$\nu \;\geq\; -\frac{1}{\lambda_i}, \quad \text{for } i = 1, \ldots, d. \tag{50}$$

For $(\mathbf{I} + \nu\Lambda)$ to be non-singular, at least one of the above inequalities will hold to equality. To ease notation, let us arrange the eigenvalues in increasing order so that

$$\lambda_1 \leq \lambda_2 \leq \cdots \leq \lambda_d.$$

For one of the equation 50 inequalities to hold to equality we need that

$$\nu \;=\; \max_{i\,:\,\lambda_i \neq 0} -\frac{1}{\lambda_i} \;=\; \max_{i\,:\,\lambda_i \neq 0} \left\{ -\frac{1}{\lambda_1}, -\frac{1}{\lambda_d} \right\}.$$

Since $(\mathbf{I} + \nu\Lambda)$ is now singular with this $\nu$, we have that the solution to equation 45 is given by

$$x \;=\; -(\mathbf{I} + \nu\Lambda)^\dagger \nu\hat{q} + n \;:=\; x^* + n, \quad \text{where } \langle x^*, n \rangle = 0, \tag{51}$$

where $\dagger$ denotes the pseudo-inverse and where $n$ is in the kernel of $(\mathbf{I} + \nu\Lambda)$, in other words $(\mathbf{I} + \nu\Lambda)n = 0$. It remains to determine $n$, which we can do with equation 46. Indeed, substituting equation 51 into equation 46 gives

$$\nabla_\nu L(x, \nu) = r + \frac{1}{2}\langle \hat{q}, x \rangle - \frac{1}{2\nu}\|x\|^2 \qquad \text{Using equation 45}$$

$$= r + \frac{1}{2}\langle \hat{q}, x^* + n \rangle - \frac{1}{2\nu}\|n\|^2 - \frac{1}{2\nu}\|x^*\|^2. \qquad \text{Using equation 51.}$$

Setting to zero and completing the squares in $n$ we have that

$$\frac{1}{2\nu}\left\| n - \frac{\nu}{2}\hat{q} \right\|^2 = r + \frac{1}{2}\langle \hat{q}, x^* \rangle - \frac{1}{2\nu}\|x^*\|^2 + \frac{\nu}{8}\|\hat{q}\|^2. \tag{52}$$

To characterize the solutions in $n$ of the above, first note that $n$ will only have a few non-zero elements. To see this, let $i^* \in \arg\max_{i\,:\,\lambda_i \neq 0} \left\{ -\frac{1}{\lambda_1}, -\frac{1}{\lambda_d} \right\}$, and note that $(\mathbf{I} + \nu\Lambda)$ has as many zeros on the diagonal as the multiplicity of the eigenvalue $\lambda_{i^*}$. That is, it has zeros elements on the indices in

$$I \;=\; \{i \;:\; \lambda_i = \lambda_{i^*}\}.$$

Thus the non-zero elements of $n$ are in the set

$$N \;=\; \{i \;:\; \lambda_i \neq \lambda_{i^*}\}.$$

Because of this observation we further re-write equation 52 as

$$\sum_{i \in N} \left( n_i - \frac{\nu}{2}\hat{q}_i \right)^2 \;=\; 2\nu r + \nu\langle \hat{q}, x^* \rangle - \|x^*\|^2 + \frac{\nu^2}{4}\|\hat{q}\|^2 - \sum_{i \in I} \frac{\nu^2}{4}\hat{q}_i^2$$

$$=\; 2\nu r + \nu\langle \hat{q}, x^* \rangle - \|x^*\|^2 + \frac{\nu^2}{4}\sum_{i \in N} \hat{q}_i^2. \tag{53}$$

Consequently, if the above is positive, then there exists solutions to the above of which

$$n_i = \frac{\nu}{2}\hat{q}_i + \frac{1}{\sqrt{|N|}}\sqrt{2\nu r + \nu\langle \hat{q}, x^* \rangle - \|x^*\|^2 + \frac{\nu^2}{4}\sum_{i \in N} \hat{q}_i^2}, \quad \text{for } i \in N. \tag{54}$$

is one. Consequently, the final solution is given by $w = z + Q(x^* + n)$ where $x^*$ is given in by equation 51. $\qquad\square$

**Corollary 1.** *If $r > 0$ and $\mathbf{P}$ has at least one negative eigenvalue, there always exists a solution to the projection equation 34.*

*Proof.* We only need to prove that there exists a solution to the quadratic equation in equation 34, after which Lemma 5 guarantees the existance of a solution. $\qquad\square$

## B   MATRIX COMPLETION

The projection equation 33 can be solved as we shown in the following theorem.

**Theorem 1.** *The solution to equation 33 is given by one of the following cases.*

1. *If $(u_j^k)^\top v_j^k = a_{i,j}$ then $u = u_i^k$ and $v = v_j^k$.*

2. *Alternatively if $u_j^k = v_j^k$ and*
$$(u_j^k)^\top v_j^k \geq 4a_{i,j}$$
   *then*
$$v = \frac{1}{2}v_j^k + \frac{1}{2}\frac{v_j^k}{\|v_j^k\|}\sqrt{\|v_j^k\|^2 - 4a_{i,j}} \tag{55}$$
$$u = -\frac{1}{2}u_j^k + \frac{1}{2}\frac{u_j^k}{\|u_j^k\|}\sqrt{\|u_j^k\|^2 - 4a_{i,j}} \tag{56}$$

3. *Finally, if none of the above holds then*
$$u = \frac{u_i^k - \gamma v_j^k}{1 - \gamma^2} \tag{57}$$
$$v = \frac{v_j^k - \gamma u_i^k}{1 - \gamma^2}, \tag{58}$$
   *where $\gamma \in (-1, 1)$ and is the solution to the depressed quartic equation*
$$(1 + \gamma^2)\langle u_i^k, v_j^k\rangle - \gamma(\|u_i^k\|^2 + \|v_j^k\|^2) = (1 - \gamma^2)^2 a_{i,j}. \tag{59}$$

*Proof.* The Lagrangian of equation 33 is given by
$$L(u, v, \gamma) = \tfrac{1}{2}\|u - u_i^k\|^2 + \tfrac{1}{2}\|v - v_j^k\|^2 + \gamma(u^\top v - a_{i,j}),$$
where $\gamma \in \mathbb{R}$ is the unknown Lagrangian Multiplier. Thus the KKT equations are
$$u - u_i^k + \gamma v = 0 \tag{60}$$
$$v - v_j^k + \gamma u = 0 \tag{61}$$
$$u^\top v = a_{i,j} \tag{62}$$

Subtracting $\gamma$ times the the second equation from the first equation, and analogously, subtracting the first equation from the second gives
$$(1 - \gamma^2)u - u_i^k + \gamma v_j^k = 0 \tag{63}$$
$$(1 - \gamma^2)v - v_j^k + \gamma u_i^k = 0. \tag{64}$$

If $\gamma = 1$, then necessarily $u_i^k = v_j^k$ and furthermore from the first equation in equation 62 we have that
$$u = u_i^k - v = v_j^k - v. \tag{65}$$
Substituting $u$ out in the original projection problem equation 33 we have that
$$\min_v \|v\|^2 - v^\top v_j^k \text{ subject to } v^\top v_j^k - \|v\|^2 = a_{i,j}. \tag{66}$$

Consequently, for every $v$ that satisfies the constraint we have that the objective value is invariant and equal to $-a_{i,j}$. Consequently there are infinite solutions. To find one such solution, we complete the squares of the constraint and find
$$\left\|v - \tfrac{1}{2}v_j^k\right\|^2 = \frac{1}{4}\|v_j^k\|^2 - a_{i,j}. \tag{67}$$

The above only has solutions if $\frac{1}{4}\|v_j^k\|^2 - a_{i,j} \geq 0$. One solution to the above is given by equation 55.

Alternatively, if $\gamma \neq 1$ then by isolating $u$ and $v$ in equation 63 and equation 64, respectively, gives

$$u = \frac{u_i^k - \gamma v_j^k}{1 - \gamma^2} \tag{68}$$

$$v = \frac{v_j^k - \gamma u_i^k}{1 - \gamma^2} \tag{69}$$

To figure out $\gamma$, we use the third constraint in equation 62 and the above two equations, which gives

$$
\begin{aligned}
u^\top v &= \frac{(u_i^k - \gamma v_j^k)^\top}{1 - \gamma^2} \frac{v_j^k - \gamma u_i^k}{1 - \gamma^2} \\
&= \frac{(1 + \gamma^2)\left\langle u_i^k, v_j^k \right\rangle - \gamma(\left\|u_i^k\right\|^2 + \left\|v_j^k\right\|^2)}{(1 - \gamma^2)^2} = a_{i,j}.
\end{aligned}
$$

Let

$$\phi(\gamma) = (1 + \gamma^2)\left\langle u_i^k, v_j^k \right\rangle - \gamma(\left\|u_i^k\right\|^2 + \left\|v_j^k\right\|^2) - (1 - \gamma^2)^2 a_{i,j}.$$

Can we now find an interval which will contain the solution in $\gamma$? Note that

$$\phi(-1) = 2\left\langle u_i^k, v_j^k \right\rangle + \left\|u_i^k\right\|^2 + \left\|v_j^k\right\|^2 = \left\|u_i^k + v_j^k\right\|^2 \geq 0$$

$$\phi(1) = 2\left\langle u_i^k, v_j^k \right\rangle - \left\|u_i^k\right\|^2 - \left\|v_j^k\right\|^2 = -\left\|u_i^k - v_j^k\right\|^2 \leq 0.$$

Thus it suffices to search for $\gamma \in (-1, \ 1)$, which can be done efficiently with bisection.

$\square$

## C PROOFS OF IMPORTANT LEMMAS

### C.1 PROOF OF LEMMA 1

Let us first describe the set of solutions for given constraint. We need to have

$$f_i + a_i x_i^T \Delta + \frac{1}{2} h_i \Delta^T x_i x_i^T \Delta = 0, \tag{70}$$

where $\Delta = w - w^t$ is unknown. If we denote by $\tau_i = x_i^T \Delta$ then equation 70 will reduce to

$$f_i + a_i \tau_i + \frac{1}{2} h_i \tau_i^2 = 0. \tag{71}$$

This quadratic equation equation 71 has solution if

$$a_i^2 - 2 h_i f_i \geq 0. \tag{72}$$

If the condition above holds then we have that the solution for $\tau$ is in this set

$$T^* := \left\{ \frac{-a_i + \sqrt{a_i^2 - 2 h_i f_i}}{h_i}, \frac{-a_i - \sqrt{a_i^2 - 2 h_i f_i}}{h_i} \right\}. \tag{73}$$

Recall that the problem equation 7 now reduces into

$$\min_\Delta \left\{ \|\Delta\|^2, \text{ such that } x_i^T \Delta \in T^* \right\}. \tag{74}$$

Note that because we want to minimize $\|\Delta\|^2$, we want to choose the constraint with smallest possible absolute value, hence the problem equation 74 is equivalent to

$$\min_\Delta \left\{ \|\Delta\|^2, \text{ such that } x_i^T \Delta = \tau_i^* \right\}, \tag{75}$$

where

$$\tau_i^* = \begin{cases} \frac{-a_i + \sqrt{a_i^2 - 2 h_i f_i}}{h_i}, & \text{if } a_i > 0 \\ \frac{-a_i - \sqrt{a_i^2 - 2 h_i f_i}}{h_i}, & \text{otherwise.} \end{cases}$$

In other words,

$$\tau_i^* = -\frac{a_i}{h_i} + \text{sign}(a_i)\frac{\sqrt{a_i^2 - 2h_i f_i}}{h_i} = \frac{a_i}{h_i}\left(\frac{\sqrt{a_i^2 - 2h_i f_i}}{|a_i|} - 1\right)$$

The final solution is hence

$$\Delta^* = \frac{\tau_i^*}{\|x_i\|^2}x_i$$

and therefore

$$w^* = w^t + \frac{\tau_i^*}{\|x_i\|^2}x_i$$

In case when equation 72 is not satisfied, and because we assumed that the loss function is non-negative, we necessary have $h_i > 0$. Then natural choice of $\tau_i$ is the one that would minimize the

$$f_i + a_i\tau_i + \frac{1}{2}h_i\tau_i^2.$$

From first order optimality conditions we obtain that

$$\tau_i^* = -\frac{a_i}{h_i}$$

which leads to equation 14.

## C.2   PROOF OF LEMMA 2

*Proof.* If $\phi(t) = 0$ then the condition holds trivially. For $t$ such that $\phi(t) \neq 0$, $\sqrt{\phi(t)}$ is differentiable, and we have

$$\frac{d^2}{dt^2}\sqrt{\phi(t)} = -\tfrac{1}{4}\phi(t)^{-3/2}\phi'(t)^2 + \tfrac{1}{2}\phi(t)^{-1/2}\phi''(t) = \tfrac{1}{4}\phi(t)^{-3/2}(-\phi'(t)^2 + 2\phi(t)\phi''(t)),$$

which is negative precisely when $\phi'(t)^2 \geq 2\phi(t)\phi''(t)$.                    □

## C.3   PROOF OF LEMMA 3

Note that

$$q(w^{t+1/2}) \overset{equation\ 17 + equation\ 15}{=} f_i(w^t) - \left\langle \nabla f_i(w^t), \frac{f_i(w^t)}{\|\nabla f_i(w^t)\|^2}\nabla f_i(w^t) \right\rangle$$

$$+ \frac{1}{2}\left\langle \nabla^2 f_i(w^t)\frac{f_i(w^t)}{\|\nabla f_i(w^t)\|^2}\nabla f_i(w^t), \frac{f_i(w^t)}{\|\nabla f_i(w^t)\|^2}\nabla f_i(w^t) \right\rangle$$

$$= \frac{1}{2}\frac{f_i(w^t)^2}{\|\nabla f_i(w^t)\|^4}\left\langle \nabla^2 f_i(w^t)\nabla f_i(w^t), \nabla f_i(w^t) \right\rangle.$$

Furthermore

$$\nabla q(w^{t+1/2}) \overset{equation\ 15}{=} \nabla f_i(w^t) + \nabla^2 f_i(w^t)(w^{t+1/2} - w^t)$$

$$\overset{equation\ 17}{=} \left(\mathbf{I} - \nabla^2 f_i(w^t)\frac{f_i(w^t)}{\|\nabla f_i(w^t)\|^2}\right)\nabla f_i(w^t).$$

Thus the second step equation 19 is given by

$$w^{t+1} = w^{t+1/2} - \frac{1}{2}\frac{f_i(w^t)^2}{\|\nabla f_i(w^t)\|^4}\frac{\left\langle \nabla^2 f_i(w^t)\nabla f_i(w^t), \nabla f_i(w^t) \right\rangle}{\left\|\left(\mathbf{I} - \nabla^2 f_i(w^t)\frac{f_i(w^t)}{\|\nabla f_i(w^t)\|^2}\right)\nabla f_i(w^t)\right\|^2} \cdot \left(\mathbf{I} - \nabla^2 f_i(w^t)\frac{f_i(w^t)}{\|\nabla f_i(w^t)\|^2}\right)\nabla f_i(w^t).$$

$$(76)$$

Putting the first equation 17 and second equation 76 updates together gives equation 20.

This gives a second order correction of the Polyak step that only requires computing a single Hessian-vector product that can be done efficiently using an additional backwards pass of the function. We call this method SP2.

## C.4 Convergence of multi-step $\mathrm{SP2}^+$

If we apply multiple steps of the $\mathrm{SP2}^+$, as opposed to two steps, the method converges to the solution of equation 15. This follows because each step of $\mathrm{SP2}^+$ is a step of NR Newton Raphson's method applied to solving the nonlinear equation

$$q(w) := f_i(w^t) + \left\langle \nabla f_i(w^t), w - w^t \right\rangle + \tfrac{1}{2} \left\langle \nabla^2 f_i(w^t)(w - w^t), w - w^t \right\rangle.$$

Indeed, starting from $w^0 = w^t$, the iterates of the NR (Newton Raphson) method are given by

$$w^{i+1} = w^i - \left( \nabla q(w^i)^\top \right)^\dagger q(w^i) = w^i - \frac{q(w^i)}{\|\nabla q(w^i)\|^2} \nabla q(w^i), \tag{77}$$

where $\mathbf{M}^\dagger$ denotes the pseudo-inverse of the matrix $\mathbf{M}$.

The NR iterates in equation 77 can also be written in a variational form given by

$$w^{i+1} = \underset{w \in \mathbb{R}^d}{\arg\min} \left\| w - w^i \right\|^2$$
$$\text{s.t. } q(w^i) + \nabla q(w^i)(w - w^i) = 0. \tag{78}$$

Comparing the above to the first equation 16 and second step equation 18 are indeed two steps of the NR method. Further, we can see that equation 78 is indeed the multi-step version of $\mathrm{SP2}^+$.

This method equation 77 is also known as gradient descent with a Polyak Step step size, or SP for short. It is this connection we will use to prove the convergence of equation 77 to a root of $q(w)$.

We assume that $q(w)$ has at least one root. Let $w_q^* \in \mathbb{R}^d$ be a least norm root of $q(w)$, that is

$$w_q^* = \arg\min \|w\|^2 \quad \text{subject to } q(w) = 0. \tag{79}$$

It follows from Theorem 3.2 of Sosa & MP Raupp (2020) that the above optimization equation 79 has solution if and only if the following matrix

$$B = (\nabla f_i(w^t) - \nabla^2 f_i(w^t)w^t)(\nabla f_i(w^t) - \nabla^2 f_i(w^t)w^t)^\top +$$
$$2 \left( -f_i(w^t) + \nabla f_i(w^t)^\top w^t - \frac{1}{2} w^{t\top} \nabla^2 f_i(w^t) w^t \right) \nabla^2 f_i(w^t) \tag{80}$$

has at least a non-negative eigenvalue.

**Theorem 2.** *Assume that the matrix $B$ defined in equation 80 has at least a non-negative eigenvalue. If $q(w)$ is star-convex with respect to $w_q^*$, that is if*

$$(w^i - w_q^*)^\top \nabla^2 f_i(w^t)(w^i - w_q^*) \geq 0, \quad \text{for all } i, \tag{81}$$

*then it follows that*

$$\min_{i=0,\dots,T-1} q(x^i) \leq \frac{\sigma_{\max}(\nabla^2 f_i(w^t))}{2T} \left\| w^0 - w_q^* \right\|^2. \tag{82}$$

*Proof.* The proof follows by applying the convergence Theorem 4.4 in Gower et al. (2020) or equivalently Corollary D.3 in Gower et al. (2021b). This result first appeared in Theorem 4.4 in Gower et al. (2020), but we apply Corollary D.3 in Gower et al. (2021b) since it is a bit simpler.

To apply this Corollary D.3 in Gower et al. (2021b), we need to verify that $q$ is an $L$–smooth function and star-convex. To verify if it is smooth, we need to find $L > 0$ such that

$$q(w) \leq q(y) + \left\langle \nabla q(y), w - y \right\rangle + \frac{L}{2} \left\| w - y \right\|^2, \tag{83}$$

which holds with $L = \sigma_{\max}(\nabla^2 q(y)) = \sigma_{\max}(\nabla^2 f_i(w^t)$ since $q$ is a quadratic function. Furthermore, for $q$ to be star-convex along the iterates $w^i$, we need to verify if

$$q(w_q^*) \geq q(w^i) + \left\langle \nabla q(w^i), w^* - w^i \right\rangle. \tag{84}$$

Since $q$ is a quadratic, we have that

$$q(w_q^*) = q(w^i) + \langle \nabla q(w^i), w^* - w^i \rangle + \langle \nabla^2 q(w^i)(w^* - w^i), w^* - w^i \rangle.$$

Using this in equation 85 gives that

$$0 \geq \langle \nabla^2 q(w^i)(w^* - w^i), w^* - w^i \rangle = \langle \nabla^2 f_i(w^t)(w^* - w^i), w^* - w^i \rangle, \tag{85}$$

which is equivalent to our assumption equation 81. We can now apply the result in Corollary D.3 in Gower et al. (2021b) which states that

$$\min_{i=0,\dots,T-1}(q(x^i) - q(w_q^*)) \leq \frac{L}{2T}\left\|w^0 - w_q^*\right\|^2.$$

Finally using $q(w_q^*) = 0$ and that $L = \sigma_{\max}(\nabla^2 f_i(w^t))$ gives the result. $\qquad\square$

To simplify notation, we will omit the dependency on $w^t$ and denote $c = f_i(w^t)$, $g = \nabla f_i(w^t)$ and $\mathbf{H} = \nabla^2 f_i(w^t)$, thus

$$q(w) = c + \langle g, w - w^t \rangle + \frac{1}{2} \langle \mathbf{H}(w - w^t), w - w^t \rangle$$
$$\nabla q(w) = g + \mathbf{H}(w - w^t)$$
$$\nabla^2 q(w) = \mathbf{H} \tag{86}$$

**Lemma 6.** *If* $g \in \mathrm{Range}(\mathbf{H})$ *and* $w^0 \in \mathrm{Range}(\mathbf{H})$ *then* $w^i, \nabla q(w^i) \in \mathrm{Range}(\mathbf{H})$ *for all* $i$ *and* $w_q^* \in \mathrm{Range}(\mathbf{H})$.

*Proof.* First, note that since $g \in \mathrm{Range}(\mathbf{H})$ and since $\nabla q(w) = q + \mathbf{H}(w - w^t)$ (see equation 86) we have that $\nabla q(w) \in \mathrm{Range}(\mathbf{H})$ for all $w$. Consequently by induction if $w^i \in \mathrm{Range}(\mathbf{H})$ then by equation 77 we have that $w^{i+1} \in \mathrm{Range}(\mathbf{H})$ since it is a combination of $\nabla q(w^i)$ and $w^i$.

Finally, let $w_q^* = w^t + w_{\mathbf{H}} + w_{\mathbf{H}}^\perp$ where $w_{\mathbf{H}} \in \mathrm{Range}(\mathbf{H})$ and $w_{\mathbf{H}}^\perp \in \mathrm{Range}(\mathbf{H})^\perp$. It follows that

$$q(w_q^*) = q(w^t + w_{\mathbf{H}}).$$

Furthermore, by orthogonality and Pythagoras' Theorem

$$\left\|w_q^*\right\| = \left\|w^t + w_{\mathbf{H}}\right\| + \left\|w_{\mathbf{H}}^\perp\right\|$$

Consequently, since $w_q^*$ is the least norm solution, we must have that $w_{\mathbf{H}}^\perp = 0$ and thus $w_q^* \in \mathrm{Range}(\mathbf{H})$. $\qquad\square$

### C.5 PROOF OF PROPOSITION 1

First we repeat the proposition for ease of reference.

**Proposition 3.** *Consider the loss functions given in equation 21. The* SP2 *method equation 7 converges according to*

$$\mathbb{E}\left\|w^{t+1} - w^*\right\|^2 \leq \rho \, \mathbb{E}\left\|w^t - w^*\right\|^2, \tag{87}$$

*where*

$$\rho = \lambda_{\max}\left(\mathbf{I} - \frac{1}{n}\sum_{i=1}^n \mathbf{H}_i \mathbf{H}_i^+\right) < 1. \tag{88}$$

*Proof.* First consider the first iterate of SP2 which applied to equation 21 are given by

$$w^{t+1} = \arg\min_{w \in \mathbb{R}^d} \left\|w - w^t\right\|^2$$
$$\text{s.t. } \|w - w^*\|_{\mathbf{H}_i}^2 = 0.$$

Thus every solution to the constraint set must satisfy

$$w \in w^* + \mathbf{N}_i \alpha, \tag{89}$$

where $\mathbf{N}_i \in \mathbb{R}^{d \times d}$ is a basis for the null space of $\mathbf{H}_i$. where $\alpha \in \mathbb{R}^d$. Substituting into the objective we have the resulting linear least squares problem given by

$$\min_{\alpha \in \mathbb{R}^d} \left\| w^* + \mathbf{N}_i \alpha - w^t \right\|^2$$

The minimal norm solution in $\alpha$ is thus

$$\alpha = \mathbf{N}_i^+ (w^t - w^*)$$

which when substituted into equation 89 gives

$$w^{t+1} \quad = \quad w^* + \mathbf{N}_i \mathbf{N}_i^+ (w^t - w^*). \tag{90}$$

Note that $\mathbf{P}_i := \mathbf{N}_i \mathbf{N}_i^+$ is the orthogonal projector onto $\mathrm{Null}(\mathbf{H}_i)$. Subtracting $w^*$ from both sides of equation 90 and applying the squared norm we have that

$$
\begin{aligned}
\left\| w^{t+1} - w^* \right\|^2 &= \left\| \mathbf{P}_i (w^t - w^*) \right\|^2 \\
&= \left\langle \mathbf{P}_i (w^t - w^*), (w^t - w^*) \right\rangle
\end{aligned} \tag{91}
$$

where we used that $\mathbf{P}_i \mathbf{P}_i = \mathbf{P}_i$ because it is a projection matrix. Now taking expectation conditioned on $w^t$ we have

$$
\begin{aligned}
\mathbb{E} \left\| w^{t+1} - w^* \right\|^2 \mid w^t &= \left\langle \mathbb{E} \mathbf{P}_i (w^t - w^*), (w^t - w^*) \right\rangle \\
&\leq \lambda_{\max} (\mathbb{E} \mathbf{P}_i) \left\| w^t - w^* \right\|^2.
\end{aligned}
$$

Since the null space is orthogonal to the range of adjoint, we have that

$$\mathbf{P}_i = \mathbf{I} - \mathbf{H}_i \mathbf{H}_i^+.$$

Thus taking expectation again gives the result equation 87.

Finally, the rate of convergence $\rho$ in equation 88 is always smaller than one because, due Jensen's inequality and that $\lambda_{\max}$ is convex over positive definite matrices we have that

$$0 < \lambda_{\max}(\mathbb{E} \mathbf{H}_i \mathbf{H}_i^*) \leq \mathbb{E} \lambda_{\max}(\mathbf{H}_i \mathbf{H}_i^*) = 1, \tag{92}$$

where the greater than zero follows since there must exist $\mathbf{H}_i \neq 0$, otherwise the result still holds and the method converges in one step (with $\rho = 0$). Now multiplying equation 92 by $-1$ then adding 1 gives

$$1 > \lambda_{\max}(\mathbf{I} - \mathbb{E} \mathbf{H}_i \mathbf{H}_i^*) \geq 0. \tag{93}$$

$$\square$$

**Remark 1** (Connection to Block Kaczmarz proof). *The proof of Proposition 3 follows the same proof technique of the Block Kaczmarz method Needell & Tropp (2014)[8]. To see this, first note that since interpolation holds, to find the minima we can solve the equations*

$$\textit{Find } w \in \mathbb{R}^d \ : \quad \nabla f_i(w) = 0, \quad \textit{for } i = 1, \dots, n. \tag{94}$$

*Since each $f_i(w)$ function is a quadratic, this is equivalent to*

$$\textit{Find } w \in \mathbb{R}^d \ : \quad \mathbf{H}_i(w - w^*) = 0, \quad \textit{for } i = 1, \dots, n. \tag{95}$$

*We can now interpret the above as one big linear system that we need to solve. If we sample one of the block of rows, say $\mathbf{H}_i(w - w^*) = 0$ for one $i$, and project $w^k$ onto this smaller system, the resulting method is the block Kaczmarz method equation 95. This also coincides with the SP2 method. Furthermore, our convergence analysis is now equivalent to applying the convergence analysis of block Kaczmarz to solving equation 95.*

---

[8]Thanks to an anonymous reviewer at ICLR for pointing out this connection.

### C.6 Proof of Proposition 2

For convenience we repeat the statement of the proposition here.

**Proposition 4.** *Consider the loss functions given in equation 21. The $SP2^+$ method equation 20 converges according to*

$$\mathbb{E}\|w^{t+1} - w^*\|^2 \leq \rho_{SP2+}^2 \, \mathbb{E}\|w^t - w^*\|^2, \tag{96}$$

*where*

$$\rho_{SP2+} = 1 - \frac{1}{2n}\sum_{i=1}^n \frac{\lambda_{\min}(\mathbf{H}_i)}{\lambda_{\max}(\mathbf{H}_i)} \tag{97}$$

*Proof.* The proof follows simply by observing that for quadratic function the $SP2^+$ is equivalent to applying two steps of the $SP$ method equation 5. Indeed in Section 3.2 the $SP2^+$ applies two steps of the $SP$ method to the local quadratic approximation of the function we wish to minimize. But in this case, since our function is quadratic, it is itself equal to it's local quadratic.

Consequently we can apply the convergence theory of $SP$ for smooth, strongly convex functions that satisfy the interpolation condition, such as Corollary 5.7.I in Gower et al. (2021b), which states that $SP$ converges at a rate of equation 97 □

### C.7 Proof of Lemma 7

The following lemma gives the two step update for $SP2L2^+$.

**Lemma 7.** *($SP2L2^+$) The $w^{t+1}$ and $s^{t+1}$ update of equation 28–equation 29 is given by*

$$w^{t+1} = w^t - (\Gamma_1 + \Gamma_2)\nabla f_i(w^t) + \Gamma_2\Gamma_1\nabla^2 f_i(w^t)\nabla f_i(w^t),$$
$$s^{t+1} = (1-\lambda)\left((1-\lambda)(s^t + \Gamma_1) + \Gamma_2\right),$$

*where* 
$$\Gamma_1 := \frac{(f_i(w^t) - (1-\lambda)s^t)_+}{1 - \lambda + \|\nabla f_i(w^t)\|^2},$$

$$\Gamma_2 := \left(\frac{f_i(w^t) - \Gamma_1\|\nabla f_i(w^t)\|^2 - (1-\lambda)^2(s^t+\Gamma_1)}{1-\lambda+\|\nabla f_i(w^t)-\Gamma_1\nabla^2 f_i(w^t)\nabla f_i(w^t)\|^2} + \frac{\frac{1}{2}\Gamma_1^2\langle\nabla^2 f_i(w^t)\nabla f_i(w^t),\nabla f_i(w^t)\rangle}{1-\lambda+\|\nabla f_i(w^t)-\Gamma_1\nabla^2 f_i(w^t)\nabla f_i(w^t)\|^2}\right)_+,$$

*where we denote* $(x)_+ = \begin{cases} x & \text{if } x \geq 0 \\ 0 & \text{otherwise} \end{cases}$.

We will use the following lemma to prove Lemma 7, which has been proven in Lemma C.2 of Gower et al. (2022).

**Lemma 8** (L2 Unidimensional Inequality Constraint). *Let $\delta > 0, c \in \mathbb{R}$ and $w, w^0, a \in \mathbb{R}^d$. The closed form solution to*

$$w', s' = \underset{w\in\mathbb{R}^d, s\in\mathbb{R}^b}{\arg\min} \; \|w - w^0\|^2 + \delta\|s - s^0\|^2$$
$$\text{s.t. } a^\top(w - w^0) + c \leq s \;, \tag{98}$$

*is given by*

$$w' = w^0 - \delta\frac{(c - s^0)_+}{1 + \delta\|a\|^2}a, \tag{99}$$

$$s' = s^0 + \frac{(c - s^0)_+}{1 + \delta\|a\|^2}, \tag{100}$$

*where we denote* $(x)_+ = \begin{cases} x & \text{if } x \geq 0 \\ 0 & \text{otherwise} \end{cases}$.

We are now in the position to prove Lemma 7. Note that

$$
\begin{aligned}
\frac{1-\lambda}{2}(s - s^t)^2 + \frac{\lambda}{2}s^2 &= \frac{1-\lambda}{2}s^2 - (1-\lambda)ss^t + \frac{\lambda}{2}s^2 + \frac{1-\lambda}{2}(s^t)^2 \\
&= \frac{1}{2}s^2 - (1-\lambda)ss^t + \frac{1-\lambda}{2}(s^t)^2 \\
&= \frac{1}{2}(s - (1-\lambda)s^t)^2 + \frac{\lambda - \lambda^2}{2}(s^t)^2.
\end{aligned}
\tag{101}
$$

Consequently equation 28 is equivalent to

$$
\begin{aligned}
w^{t+1/2}, s^{t+1/2} \quad &= \quad \underset{s \geq 0,\ w \in \mathbb{R}^d}{\arg\min} \left\| w - w^t \right\|^2 + \frac{1}{1-\lambda}(s - (1-\lambda)s^t)^2 \\
&\text{s.t. } q_{i,t}(w^t) + \left\langle \nabla q_{i,t}(w^t), w - w^t \right\rangle \leq s.
\end{aligned}
\tag{102}
$$

It follows from Lemma 8 that the closed form solution is

$$
w^{t+1/2} = w^t - \frac{1}{1-\lambda}\frac{(q_{i,t}(w^t) - (1-\lambda)s^t)_+}{1 + \frac{1}{1-\lambda}\left\| \nabla q_{i,t}(w^t) \right\|^2}\nabla q_{i,t}(w^t),
\tag{103}
$$

$$
s^{t+1/2} = (1-\lambda)s^t + \frac{(q_{i,t}(w^t) - (1-\lambda)s^t)_+}{1 + \frac{1}{1-\lambda}\left\| \nabla q_{i,t}(w^t) \right\|^2},
\tag{104}
$$

where we denote

$$
(x)_+ = \begin{cases} x & \text{if } x \geq 0 \\ 0 & \text{otherwise} \end{cases}.
$$

Note that $q_{i,t}(w^t) = f_i(w^t)$ and $\nabla q_{i,t}(w^t) = \nabla f_i(w^t)$. To simplify the notation, we also denote

$$
\Gamma_1 = \frac{1}{1-\lambda}\frac{(f_i(w^t) - (1-\lambda)s^t)_+}{1 + \frac{1}{1-\lambda}\left\| \nabla f_i(w^t) \right\|^2}.
$$

With this notation we have that

$$
w^{t+1/2} = w^t - \frac{1}{1-\lambda}\frac{(f_i(w^t) - (1-\lambda)s^t)_+}{1 + \frac{1}{1-\lambda}\left\| \nabla f_i(w^t) \right\|^2}\nabla f_i(w^t)
\tag{105}
$$

$$
= w^t - \Gamma_1 \nabla f_i(w^t),
\tag{106}
$$

$$
s^{t+1/2} = (1-\lambda)s^t + \frac{(f_i(w^t) - (1-\lambda)s^t)_+}{1 + \frac{1}{1-\lambda}\left\| \nabla f_i(w^t) \right\|^2}
\tag{107}
$$

$$
= (1-\lambda)(s^t + \Gamma_1).
\tag{108}
$$

In a completely analogous way, the closed form solution to equation 29 is

$$
w^{t+1} = w^{t+1/2} - \frac{1}{1-\lambda}\frac{(q_{i,t}(w^{t+1/2}) - (1-\lambda)s^{t+1/2})_+}{1 + \frac{1}{1-\lambda}\left\| \nabla q_{i,t}(w^{t+1/2}) \right\|^2} \cdot \nabla q_{i,t}(w^{t+1/2}),
$$

$$
s^{t+1} = (1-\lambda)s^{t+1/2} + \frac{(q_{i,t}(w^{t+1/2}) - (1-\lambda)s^{t+1/2})_+}{1 + \frac{1}{1-\lambda}\left\| \nabla q_{i,t}(w^{t+1/2}) \right\|^2}.
\tag{109}
$$

Note that

$$
\begin{aligned}
q_{i,t}(w^{t+1/2}) &= f_i(w^t) - \left\langle \nabla f_i(w^t), \Gamma_1 \nabla f_i(w^t) \right\rangle + \frac{1}{2}\left\langle \nabla^2 f_i(w^t)\Gamma_1 \nabla f_i(w^t), \Gamma_1 \nabla f_i(w^t) \right\rangle \\
&= f_i(w^t) - \Gamma_1 \left\| \nabla f_i(w^t) \right\|^2 + \frac{1}{2}\Gamma_1^2 \left\langle \nabla^2 f_i(w^t)\nabla f_i(w^t), \nabla f_i(w^t) \right\rangle
\end{aligned}
$$

and

$$
\begin{aligned}
\nabla q_{i,t}(w^{t+1/2}) &= \nabla f_i(w^t) + \nabla^2 f_i(w^t)(w^{t+1/2} - w^t) \\
&= \nabla f_i(w^t) - \Gamma_1 \nabla^2 f_i(w^t)\nabla f_i(w^t).
\end{aligned}
$$

Denoting $\Gamma_2$ as in the statement of the lemma we conclude that

$$
\begin{aligned}
w^{t+1} &= w^{t+1/2} - \frac{1}{1-\lambda} \frac{(q_{i,t}(w^{t+1/2}) - (1-\lambda)s^{t+1/2})_+}{1 + \frac{1}{1-\lambda}\left\|\nabla q_{i,t}(w^{t+1/2})\right\|^2} \cdot \nabla q_{i,t}(w^{t+1/2}) \\
&= w^t - \Gamma_1 \nabla f_i(w^t) - \Gamma_2 \left(\nabla f_i(w^t) - \Gamma_1 \nabla^2 f_i(w^t) \nabla f_i(w^t)\right),
\end{aligned}
\tag{110}
$$

$$
\begin{aligned}
s^{t+1} &= (1-\lambda)s^{t+1/2} + \frac{(q_{i,t}(w^{t+1/2}) - (1-\lambda)s^{t+1/2})_+}{1 + \frac{1}{1-\lambda}\left\|\nabla q_{i,t}(w^{t+1/2})\right\|^2} \\
&= (1-\lambda)\left((1-\lambda)(s^t + \Gamma_1) + \Gamma_2\right)
\end{aligned}
\tag{111}
$$

## C.8 PROOF OF LEMMA 9

The following Lemma gives a closed form for the two-step update for $\texttt{SP2L1}^+$.

**Lemma 9.** *($\texttt{SP2L1}^+$) The $w^{t+1}$ and $s^{t+1}$ update is given by*

$$
w^{t+1} = w^t - (\Gamma_4 + \Gamma_6)\nabla f_i(w^t) + \Gamma_6 \Gamma_4 \nabla^2 f_i(w^t)\nabla f_i(w^t),
$$

$$
s^{t+1} = \left(\left(s^t - \frac{\lambda}{2(1-\lambda)} + \Gamma_3\right)_+ - \frac{\lambda}{2(1-\lambda)} + \Gamma_5\right)_+,
$$

*where*

$$
\Gamma_3 = \frac{\left(f_i(w^t) - \left(s^t - \frac{\lambda}{2(1-\lambda)}\right)\right)_+}{1 + \|\nabla f_i(w^t)\|^2}, \quad \Gamma_4 = \min\left\{\Gamma_3, \frac{f_i(w^t)}{\|\nabla f_i(w^t)\|^2}\right\},
$$

$$
\Gamma_5 = \frac{\left(\Lambda_1 - \left(s^t - \frac{\lambda}{2(1-\lambda)}\right)\right)_+}{1 + \|\nabla f_i(w^t) - \Gamma_4 \nabla^2 f_i(w^t)\nabla f_i(w^t)\|^2},
$$

$$
\Gamma_6 = \min\left\{\Gamma_5, \frac{\Lambda_1}{\|\nabla f_i(w^t) - \Gamma_4 \nabla^2 f_i(w^t)\nabla f_i(w^t)\|^2}\right\},
$$

$$
\Lambda_1 = f_i(w^t) - \Gamma_4 \left\|\nabla f_i(w^t)\right\|^2 + \frac{1}{2}\Gamma_4^2 \left\langle \nabla^2 f_i(w^t)\nabla f_i(w^t), \nabla f_i(w^t)\right\rangle.
$$

To solve equation 30, we consider the following two-step method similar to equation 28 and equation 29:

$$
w^{t+1/2}, s^{t+1/2} = \underset{s \geq 0,\, w \in \mathbb{R}^d}{\arg\min} \frac{1-\lambda}{2}\Delta_t + \frac{\lambda}{2}s
\tag{112}
$$

$$
\text{s.t. } q_{i,t}(w^t) + \left\langle \nabla q_{i,t}(w^t), w - w^t\right\rangle \leq s.
$$

$$
w^{t+1}, s^{t+1} = \underset{s \geq 0,\, w \in \mathbb{R}^d}{\arg\min} \frac{1-\lambda}{2}\Delta_{t+\frac{1}{2}} + \frac{\lambda}{2}s
\tag{113}
$$

$$
\text{s.t. } q_{i,t}(w^{t+1/2}) + \left\langle \nabla q_{i,t}(w^{t+1/2}), w - w^{t+1/2}\right\rangle \leq s.
$$

Note that

$$
\frac{1-\lambda}{2}(s - s^t)^2 + \frac{\lambda}{2}s = \frac{1-\lambda}{2}\left(s - \left(s^t - \frac{\lambda}{2(1-\lambda)}\right)\right)^2 + \text{constants w.r.t. } w \text{ and } s.
$$

Then, equation 112 is equivalent to solving

$$
w^{t+1/2}, s^{t+1/2} = \underset{s \geq 0,\, w \in \mathbb{R}^d}{\arg\min} \|w - w^t\|^2 + \left(s - \left(s^t - \frac{\lambda}{2(1-\lambda)}\right)\right)^2
\tag{114}
$$

$$
\text{s.t. } q_{i,t}(w^t) + \left\langle \nabla q_{i,t}(w^t), w - w^t\right\rangle \leq s.
$$

It follows from Lemma C.4 in Gower et al. (2022) that the closed form solution to equation 112 is

$$w^{t+1/2} = w^t - \min\left\{\frac{\left(q_{i,t}(w^t) - \left(s^t - \frac{\lambda}{2(1-\lambda)}\right)\right)_+}{1 + \|\nabla q_{i,t}(w^t)\|^2}, \frac{q_{i,t}(w^t)}{\|\nabla q_{i,t}(w^t)\|^2}\right\}\nabla q_{i,t}(w^t),$$

$$s^{t+1/2} = \left(\left(s^t - \frac{\lambda}{2(1-\lambda)}\right) + \frac{\left(q_{i,t}(w^t) - \left(s^t - \frac{\lambda}{2(1-\lambda)}\right)\right)_+}{1 + \|\nabla q_{i,t}(w^t)\|^2}\right)_+,$$

where we denote $(x)_+ = \begin{cases} x & \text{if } x \geq 0 \\ 0 & \text{otherwise} \end{cases}$.

Note that $q_{i,t}(w^t) = f_i(w^t)$ and $\nabla q_{i,t}(w^t) = \nabla f_i(w^t)$. To simplify the notation, denote

$$\Gamma_3 = \frac{\left(f_i(w^t) - \left(s^t - \frac{\lambda}{2(1-\lambda)}\right)\right)_+}{1 + \|\nabla f_i(w^t)\|^2},$$

and

$$\Gamma_4 = \min\left\{\Gamma_3, \frac{f_i(w^t)}{\|\nabla f_i(w^t)\|^2}\right\}.$$

Then, we have

$$w^{t+1/2} = w^t - \Gamma_4 \nabla f_i(w^t),$$
$$s^{t+1/2} = \left(\left(s^t - \frac{\lambda}{2(1-\lambda)}\right) + \Gamma_3\right)_+.$$

In a similar way, we can get the closed form solution to equation 113, which is given as

$$w^{t+1} = w^{t+1/2} - \min\left\{\frac{\left(q_{i,t}(w^{t+1/2}) - \left(s^{t+1/2} - \frac{\lambda}{2(1-\lambda)}\right)\right)_+}{1 + \|\nabla q_{i,t}(w^{t+1/2})\|^2}, \frac{q_{i,t}(w^{t+1/2})}{\|\nabla q_{i,t}(w^{t+1/2})\|^2}\right\}\nabla q_{i,t}(w^{t+1/2}),$$

$$s^{t+1} = \left(\left(s^{t+1/2} - \frac{\lambda}{2(1-\lambda)}\right) + \frac{\left(q_{i,t}(w^{t+1/2}) - \left(s^t - \frac{\lambda}{2(1-\lambda)}\right)\right)_+}{1 + \|\nabla q_{i,t}(w^{t+1/2})\|^2}\right)_+.$$

Note that

$$q_{i,t}(w^{t+1/2}) = f_i(w^t) - \left\langle\nabla f_i(w^t), \Gamma_4 \nabla f_i(w^t)\right\rangle + \frac{1}{2}\left\langle\nabla^2 f_i(w^t)\Gamma_4\nabla f_i(w^t), \Gamma_4\nabla f_i(w^t)\right\rangle$$

$$= f_i(w^t) - \Gamma_4\left\|\nabla f_i(w^t)\right\|^2 + \frac{1}{2}\Gamma_4^2\left\langle\nabla^2 f_i(w^t)\nabla f_i(w^t), \nabla f_i(w^t)\right\rangle$$

$$\triangleq \Lambda_1$$

and

$$\nabla q_{i,t}(w^{t+1/2}) = \nabla f_i(w^t) + \nabla^2 f_i(w^t)(w^{t+1/2} - w^t)$$
$$= \nabla f_i(w^t) - \Gamma_4\nabla^2 f_i(w^t)\nabla f_i(w^t).$$

Again, to simplify the notation, we denote

$$\Gamma_5 = \frac{\left(q_{i,t}(w^{t+1/2}) - \left(s^t - \frac{\lambda}{2(1-\lambda)}\right)\right)_+}{1 + \|\nabla q_{i,t}(w^{t+1/2})\|^2}$$

$$= \frac{\left(\Lambda_1 - \left(s^t - \frac{\lambda}{2(1-\lambda)}\right)\right)_+}{1 + \|\nabla f_i(w^t) - \Gamma_4\nabla^2 f_i(w^t)\nabla f_i(w^t)\|^2},$$

and

$$\Gamma_6 = \min \left\{ \Gamma_5, \frac{q_{i,t}(w^{t+1/2})}{\|\nabla q_{i,t}(w^{t+1/2})\|^2} \right\}$$
$$= \min \left\{ \Gamma_5, \frac{\Lambda_1}{\|\nabla f_i(w^t) - \Gamma_4 \nabla^2 f_i(w^t) \nabla f_i(w^t)\|^2} \right\}.$$

Then, we have

$$w^{t+1} = w^{t+1/2} - \Gamma_6 \nabla q_{i,t}(w^{t+1/2})$$
$$= w^t - \Gamma_4 \nabla f_i(w^t) - \Gamma_6 \left( \nabla f_i(w^t) - \Gamma_4 \nabla^2 f_i(w^t) \nabla f_i(w^t) \right)$$
$$= w^t - (\Gamma_4 + \Gamma_6) \nabla f_i(w^t) + \Gamma_6 \Gamma_4 \nabla^2 f_i(w^t) \nabla f_i(w^t),$$

$$s^{t+1} = \left( \left( s^{t+1/2} - \frac{\lambda}{2(1-\lambda)} \right) + \Gamma_5 \right)_+$$
$$= \left( \left( \left( \left( s^t - \frac{\lambda}{2(1-\lambda)} \right) + \Gamma_3 \right)_+ - \frac{\lambda}{2(1-\lambda)} \right) + \Gamma_5 \right)_+.$$

## C.9 Proof of Lemma 10

The following lemma gives a closed form for two step method $\texttt{SP2max}^+$.

**Lemma 10.** ($\texttt{SP2max}^+$) *The $w^{t+1}$ and $s^{t+1}$ update is given by*

$$w^{t+1} = w^t - (\Gamma_1 + \Gamma_3) \nabla f_i(w^t) + \Gamma_3 \Gamma_1 \nabla^2 f_i(w^t) \nabla f_i(w^t),$$
$$s^{t+1} = \max \left\{ \Gamma_2 - \frac{\lambda}{2(1-\lambda)} \left\| \nabla f_i(w^t) - \Gamma_1 \nabla^2 f_i(w^t) \nabla f_i(w^t) \right\|^2, 0 \right\},$$

*where*

$$\Gamma_1 = \min \left\{ \frac{f_i(w^t)}{\|\nabla f_i(w^t)\|^2}, \frac{\lambda}{2(1-\lambda)} \right\},$$
$$\Gamma_2 = f_i(w^t) - \Gamma_1 \left\| \nabla f_i(w^t) \right\|^2 + \tfrac{1}{2} \Gamma_1^2 \left\langle \nabla^2 f_i(w^t) \nabla f_i(w^t), \nabla f_i(w^t) \right\rangle,$$
$$\Gamma_3 = \min \left\{ \frac{\Gamma_2}{\|\nabla f_i(w^t) - \Gamma_1 \nabla^2 f_i(w^t) \nabla f_i(w^t)\|^2}, \frac{\lambda}{2(1-\lambda)} \right\}.$$

To solve equation 31, we again consider a two step method similar to equation 112 and equation 113:

$$w^{t+1/2}, s^{t+1/2} = \operatorname*{arg\,min}_{s \geq 0, \, w \in \mathbb{R}^d} \frac{1-\lambda}{2} \left\| w - w^t \right\|^2 + \frac{\lambda}{2} s \tag{115}$$
$$\text{s.t. } q_{i,t}(w^t) + \left\langle \nabla q_{i,t}(w^t), w - w^t \right\rangle \leq s.$$

$$w^{t+1}, s^{t+1} = \operatorname*{arg\,min}_{s \geq 0, \, w \in \mathbb{R}^d} \frac{1-\lambda}{2} \left\| w - w^{t+1/2} \right\|^2 + \frac{\lambda}{2} s \tag{116}$$
$$\text{s.t. } q_{i,t}(w^{t+1/2}) + \left\langle \nabla q_{i,t}(w^{t+1/2}), w - w^{t+1/2} \right\rangle \leq s.$$

Note that equation 115 is equivalent to solving

$$w^{t+1/2}, s^{t+1/2} = \operatorname*{arg\,min}_{s \geq 0, \, w \in \mathbb{R}^d} \frac{1}{2} \left\| w - w^t \right\|^2 + \frac{\lambda}{2(1-\lambda)} s \tag{117}$$
$$\text{s.t. } q_{i,t}(w^t) + \left\langle \nabla q_{i,t}(w^t), w - w^t \right\rangle \leq s.$$

It follows from Lemma D.2 in Gower et al. (2022) that the closed form solution to equation 117 is

$$w^{t+1/2} = w^t - \min\left\{\frac{q_{i,t}(w^t)}{\|\nabla q_{i,t}(w^t)\|^2}, \frac{\lambda}{2(1-\lambda)}\right\}\nabla q_{i,t}(w^t)$$

$$= w^t - \min\left\{\frac{f_i(w^t)}{\|\nabla f_i(w^t)\|^2}, \frac{\lambda}{2(1-\lambda)}\right\}\nabla f_i(w^t)$$

$$= w_t - \Sigma_1 \nabla f_i(w^t),$$

$$s^{t+1/2} = \max\left\{q_{i,t}(w^t) - \frac{\lambda}{2(1-\lambda)}\|\nabla q_{i,t}(w^t)\|^2, 0\right\}$$

$$= \max\left\{f_i(w^t) - \frac{\lambda}{2(1-\lambda)}\|\nabla f_i(w^t)\|^2, 0\right\},$$

where we denote

$$\Gamma_1 = \min\left\{\frac{f_i(w^t)}{\|\nabla f_i(w^t)\|^2}, \frac{\lambda}{2(1-\lambda)}\right\}.$$

Note that

$$q_{i,t}(w^{t+1/2}) = f_i(w^t) - \Gamma_1\|\nabla f_i(w^t)\|^2 + \tfrac{1}{2}\Gamma_1^2\langle\nabla^2 f_i(w^t)\nabla f_i(w^t), \nabla f_i(w^t)\rangle$$

$$:= \Gamma_2,$$

and

$$q_{i,t}(w^{t+1/2}) = \nabla f_i(w^t) + \nabla^2 f_i(w^t)(w^{t+1/2} - w^t)$$

$$= \nabla f_i(w^t) - \Gamma_1\nabla^2 f_i(w^t)\nabla f_i(w^t).$$

Similarly, we have the closed form solution to equation 116 given as

$$w^{t+1} = w^{t+1/2} - \min\left\{\frac{q_{i,t}(w^{t+1/2})}{\|\nabla q_{i,t}(w^{t+1/2})\|^2}, \frac{\lambda}{2(1-\lambda)}\right\}\nabla q_{i,t}(w^{t+1/2})$$

$$= w^{t+1/2} - \min\left\{\frac{\Gamma_2}{\|\nabla f_i(w^t) - \Gamma_1\nabla^2 f_i(w^t)\nabla f_i(w^t)\|^2}, \frac{\lambda}{2(1-\lambda)}\right\}(\nabla f_i(w^t) - \Gamma_1\nabla^2 f_i(w^t)\nabla f_i(w^t))$$

$$= w^{t+1/2} - \Gamma_3\left(\nabla f_i(w^t) - \Gamma_1\nabla^2 f_i(w^t)\nabla f_i(w^t)\right)$$

$$= w^t - (\Gamma_1 + \Gamma_3)\nabla f_i(w^t) + \Gamma_3\Gamma_1\nabla^2 f_i(w^t)\nabla f_i(w^t)$$

$$s^{t+1} = \max\left\{q_{i,t}(w^{t+1/2}) - \frac{\lambda}{2(1-\lambda)}\|\nabla q_{i,t}(w^{t+1/2})\|^2, 0\right\}$$

$$= \max\left\{\Gamma_2 - \frac{\lambda}{2(1-\lambda)}\|\nabla f_i(w^t) - \Gamma_1\nabla^2 f_i(w^t)\nabla f_i(w^t)\|^2, 0\right\}.$$

### C.10 PROOF OF LEMMA 4

In GLMs, the unregularized problem equation 31 becomes

$$w^{t+1}, s^{t+1} = \arg\min_{s\geq 0,\, w\in\mathbb{R}^d} \frac{1}{2}\|w - w^t\|^2 + \widetilde{\lambda}s$$

$$\text{s.t. } f_i + \langle a_i x_i, w - w^t\rangle + \tfrac{1}{2}\langle h_i x_i x_i^\top(w - w^t), w - w^t\rangle \leq s, \qquad (118)$$

where $\widetilde{\lambda} := \frac{\lambda}{2(1-\lambda)}$, and we denote $f_i = f_i(w^t)$, $a_i := \phi_i'(x_i^\top w - y_i)$, $h_i := \phi_i''(x_i^\top w - y_i)$ for short.

Denote $\triangle := w - w^t$. Then, problem equation 118 reduces to

$$\min_{s\geq 0,\, \triangle\in\mathbb{R}^d} \frac{1}{2}\|\triangle\|^2 + \widetilde{\lambda}s$$

$$\text{s.t. } f_i + a_i x_i^\top\triangle + \tfrac{1}{2}h_i\triangle^\top x_i x_i^\top\triangle \leq s. \qquad (119)$$

Note that we want to minimize $\|\triangle\|^2$. Together with the above constraint, we can conclude that $\triangle$ must be a multiple of $x_i$ since any other component will not help satisfy the constraint but increase $\|\triangle\|^2$. Let $\triangle = cx_i$ and $\ell = \|x_i\|^2$, then problem equation 119 becomes

$$\min_{s \geq 0,\, c \in \mathbb{R}} \frac{1}{2}c^2\ell + \widetilde{\lambda}s$$
$$\text{s.t. } f_i + a_i\ell c + \frac{1}{2}h_i\ell^2 c^2 \leq s. \tag{120}$$

The corresponding Lagrangian function is then given as

$$L(s, c, \nu_1, \nu_2) = \frac{1}{2}c^2\ell + \widetilde{\lambda}s + \nu_1(f_i + a_i\ell c + \frac{1}{2}h_i\ell^2 c^2 - s) - \nu_2 s,$$

where $\nu_1, \nu_2 \geq 0$ are the Lagrangian multipliers. The KKT conditions are thus

$$f_i + a_i\ell c + \frac{1}{2}h_i\ell^2 c^2 - s \leq 0,$$
$$s \geq 0, \quad \nu_1 \geq 0, \quad \nu_2 \geq 0,$$
$$\nu_1(f_i + a_i\ell c + \frac{1}{2}h_i\ell^2 c^2 - s) = 0, \quad \nu_2 s = 0,$$
$$\widetilde{\lambda} - \nu_1 - \nu_2 = 0,$$
$$\ell c + \nu_1 a_i\ell + \nu_1 h_i\ell^2 c = 0.$$

By checking the complementary conditions, the solution to the above KKT equations has three cases, which are summarized below.

Case I: The Lagragian multiplier $\nu_2 = 0$. In which case $\nu_1^\star = \widetilde{\lambda}$, $\nu_2^\star = 0$, $c^\star = -\frac{\widetilde{\lambda}a_i}{1+\widetilde{\lambda}h_i\ell}$, and

$$s^\star = f_i + a_i\ell c^\star + \frac{1}{2}h_i\ell^2 c^{\star 2} = f_i - \frac{\widetilde{\lambda}a_i^2\ell}{1 + \widetilde{\lambda}h_i\ell} + \frac{h_i\widetilde{\lambda}^2 a_i^2\ell^2}{2(1 + \widetilde{\lambda}h_i\ell)^2},$$

which is feasible if $s^\star \geq 0$. The resulting objective function is $\frac{1}{2}c^{\star 2}\ell + \widetilde{\lambda}s^\star$, which is $\geq 0$.

Case II: The Lagragian multiplier $\nu_1 = 0$. In which case $\nu_1^\star = 0$, $\nu_2^\star = \widetilde{\lambda}$, $c^\star = 0$, $s^\star = 0$, which is feasible if $f_i = 0$. The objective function is 0 in this case and the variable $w$ is unchanged since $w - w^t = c^\star x_i = 0$.

Case III: Neither Lagragian multiplier is zero. In which case there are two possible solutions for $c$ given by $c^\star = \frac{-a_i \pm \sqrt{a_i^2 - 2h_i f_i}}{h_i\ell}$, $\nu_1^\star = -\frac{c}{a_i + h_i\ell c}$, $\nu_2^\star = \widetilde{\lambda} + \frac{c}{a_i + h_i\ell c}$, $s^\star = 0$. Note that

$$a_i + h_i\ell c = \pm\sqrt{a_i^2 - 2h_i f_i}.$$

Consequently to guarantee that the Lagrangian multipliers $\nu_1$ and $\nu_2$ are non-negative, we must have $c^\star = \frac{-a_i + \sqrt{a_i^2 - 2h_i f_i}}{h_i\ell}$ and in this case the objective function equals $\frac{1}{2}c^{\star 2}\ell \geq 0$.

As a summary, if $f_i = 0$, Case II is the optimal solution. Alternatively if $f_i > 0$ and if

$$\widetilde{s} = f_i - \frac{\widetilde{\lambda}a_i^2\ell}{1 + \widetilde{\lambda}h_i\ell} + \frac{h_i\widetilde{\lambda}^2 a_i^2\ell^2}{2(1 + \widetilde{\lambda}h_i\ell)^2}$$

is non-negative then Case I is the optimal solution. Otherwise, Case III with $c^\star = \frac{-a_i + \sqrt{a_i^2 - 2h_i f_i}}{h_i\ell}$ is the optimal solution.

Therefore, the optimal solution to equation 118 is then

$$w^{t+1} = w^t + c^\star x_i,$$
$$s^{t+1} = \max\{\widetilde{s},\, 0\},$$

where

$$c^\star = \begin{cases} 0, & \text{if } f_i = 0 \\ -\frac{\widetilde{\lambda}a_i}{1+\widetilde{\lambda}h_i\ell}, & \text{if } f_i > 0 \text{ and } \widetilde{s} \geq 0, \\ \frac{-a_i + \sqrt{a_i^2 - 2h_i f_i}}{h_i\ell}, & \text{otherwise.} \end{cases}$$

# D ADDITIONAL NUMERICAL EXPERIMENTS

## D.1 NON-CONVEX PROBLEMS

For the non-convex experiements, we used the Python Package `pybenchfunction` available on github `Python_Benchmark_Test_Optimization_Function_Single_Objective`.

We compared `SP2` equation 7 (blue curve) , `SP2`$^+$ equation 20 (green curve) on the non-convex problems PermD$\beta^+$, Rastrigin and Levy N. 13, Rosenbrock Jamil & Yang (2013)[9]. For a baseline we compared against `SGD` (yellow ), Adam (pink) and `Newton` (red).

For the implementation of `Newton`'s method, we found that the regularized Newton methods was best suited for these non-convex experiments where

$$w^{t+1} = w^t - \gamma(\mathbf{I}\epsilon + \nabla^2 f(w^t))^{-1}\nabla f(w^t),$$

where $\epsilon$ is the regularization parameter which we set to $\epsilon = 10^{-8}$, and $\gamma$ is a learning rate which we tuned.

Specifically, for all the experiments in this section we set a learning for all methods to $0.25$ except for SGD and ADAM, both which had to be tuned by doing a grid search on the first 100 epochs over the grid

$$[0.0001, 0.0005, 0.001, 0.005, 0.01, , 0.05, 0.25, 1]$$

proving much harder to tune.

### D.1.1 PERMD$\beta^+$ IS AN INCORRECT IMPLEMENTATION OF PERMD$\beta$.

We note here that the PermD$\beta^+$ implemented is given by

$$\texttt{PermD}\beta^+(x) := \sum_{i=1}^d \sum_{j=1}^d \left( (j^i + \beta) \left( \left(\frac{x_j}{j}\right)^i - 1 \right) \right)^2.$$

which is different than the standard `PermD`$\beta$ function which is given by

$$\texttt{PermD}\beta(x) := \sum_{i=1}^d \left( \sum_{j=1}^d (j^i + \beta) \left( \left(\frac{x_j}{j}\right)^i - 1 \right) \right)^2.$$

We believe this is a small mistake, which is why we have introduced the plus in `PermD`$\beta^+$ to distinguish this function from the standard `PermD`$\beta$ function. Yet, the `PermD`$\beta^+$ is still an interesting non-convex problem, and thus we have used it in our experiments despite this small alteration.

### D.1.2 THE LEVY N. 13 AND ROSENBROCK PROBLEMS

Here we provide two additional experiments on the non-convex function Levy N. 13 and Rosenbrock that complement the findings in 6.1.

For the Levy N. 13 function in Figure 6 we have that again `SP2` converges in 10 epochs to the global minima. In contrast Newton's method converges immediately to a local maxima, that can be easily seen on the surface plot of the right of Figure 6.

The one problem where `SP2` was not the fastest was on the Rosenbrock function, see Figure 7. Here `Newton` was equally fast. But note, this problem was designed to emphasize the advantages of Newton over gradient descent. For completeness, we also give the function value versus time plot in Figure 8.

---

[9]We used the Python Package `pybenchfunction` available on github `Python_Benchmark_Test_Optimization_Function_Single_Objective`. We also note that the PermD$\beta^+$ implemented in this package is a modified version of the `PermD`$\beta$ function, as we detail in Section D.1.1.

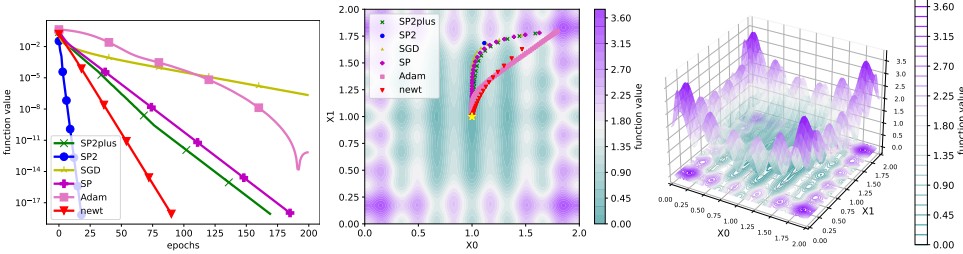

Figure 6: The Levy N. 13 function where Left: we plot $f(x)$ across epochs Middle: level set plot, Right: Surface plot. SP2 is in blue , SP2$^+$ is in green , SGD is in yellow and Newton is in red .

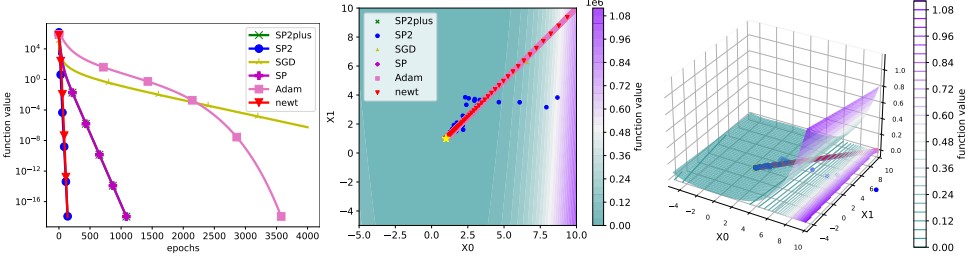

Figure 7: The Rosenbrock function where Left: we plot $f(x)$ across epochs Middle: level set plot, Right: Surface plot. SP2 is in blue , SP2$^+$ is in green , SGD is in yellow and Newton is in red .

### D.1.3 LARGER DIMENSIONS

Here we increase the dimension of the problems to appraise the effect dimensions has on the convergence of the methods, see Figure 9 for a comparison in terms of epochs, and Figure 3 in the main text for a comparison in terms of time taken. We no longer compare to Newton's method because it was exceeding our maximum allocated time. We find that when increasing the dimension, these problems become extremely challenging. To select the learning rate of each method, we used a grid search over the first 100 epochs. For the SP2, SP2$^+$ and SP methods we used the grid

$$[0.005, 0.01, 0.05, 0.1].$$

For Adam and SGD we used the larger grid

$$[0.00001, 0.00005, 0.0001, 0.005, 0.01, 0.05, 0.1].$$

Thus we gave a competitive advantage to SGD and Adam.

### D.2 ADDITIONAL CONVEX EXPERIMENTS

We set the desired tolerance for each algorithm to $0.01$, and set the maximum number of epochs for each algorithm to 200 in colon-cancer and 30 in mushrooms. To choose an optimal slack parameter $\lambda$ for SP2L2$^+$, SP2L1$^+$, and SP2max$^+$, we test these three methods on a uniform grid $\lambda \in \{0.1, 0.2, \ldots, 0.9\}$ with $\sigma = 0.001$. The gradient norm and loss evaluated at each epoch are presented in Figures 10-13 (see Appendix D). It can be seen that SP2L2$^+$ performs best when $\lambda = 0.9$ in colon-cancer and $\lambda = 0.1$ in mushrooms, SP2L1$^+$ and SP2max$^+$ perform best when $\lambda = 0.1$ in both data sets. Therefore, we set $\lambda = 0.9$ for SP2L2$^+$ in colon-cancer and fix $\lambda = 0.1$ in other cases.

Under the same setting as in Section 6.3, we also compare the SP2max and SP2max$^+$ methods on a grid $\lambda = [0.001 \ 0.01 : 0.01 : 0.05]$ with $\sigma = 0$. The gradient norm and loss evaluated at each epoch are presented in Figures 14-15 (see Appendix D). As we observe, SP2max$^+$ always outperforms the SP2max method.

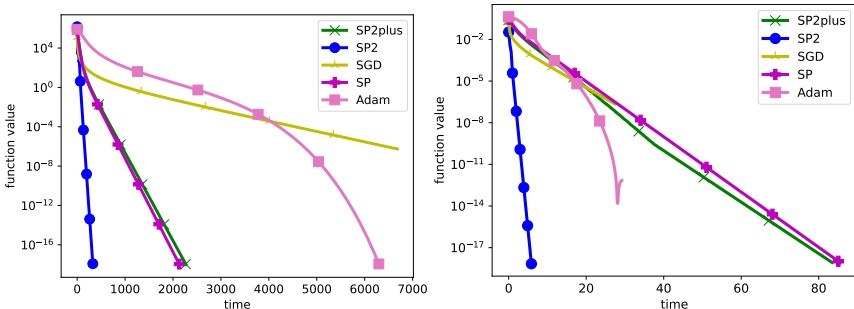

Figure 8: The time plots for the Rosenbrock function (Left) and Levy N. 13 function (Right). SP2 is in blue , SP2$^+$ is in green , SGD is in yellow , Adam and is in pink.

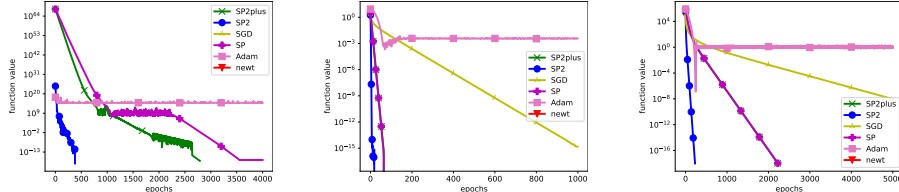

Figure 9: Function value vs epochs for the PermD$\beta^+$ ($d = 10$) function with $\beta = 0.5$ (Left), the Rastrigin function ($d = 100$) (Middle) and the Rosenbrock function ($d = 100$).

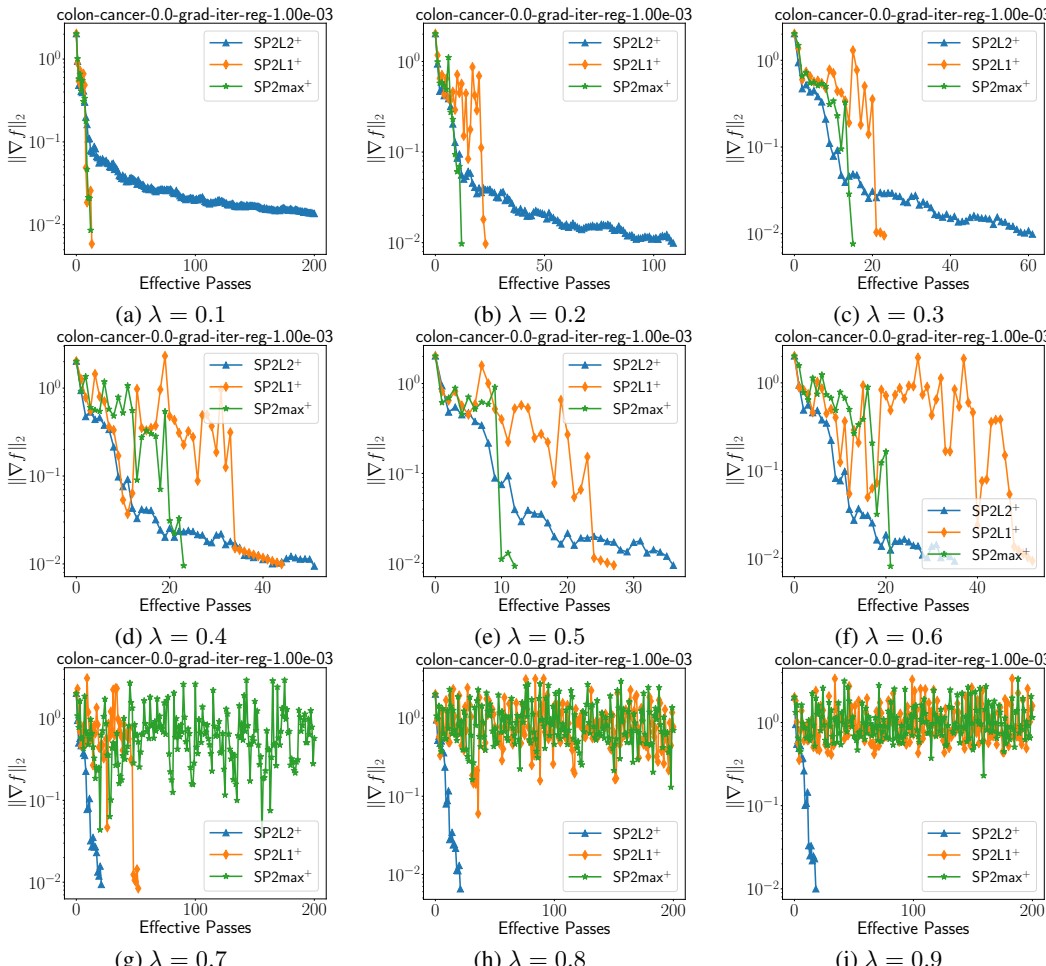

Figure 10: Colon-cancer: gradient norm at each epoch with different $\lambda$.

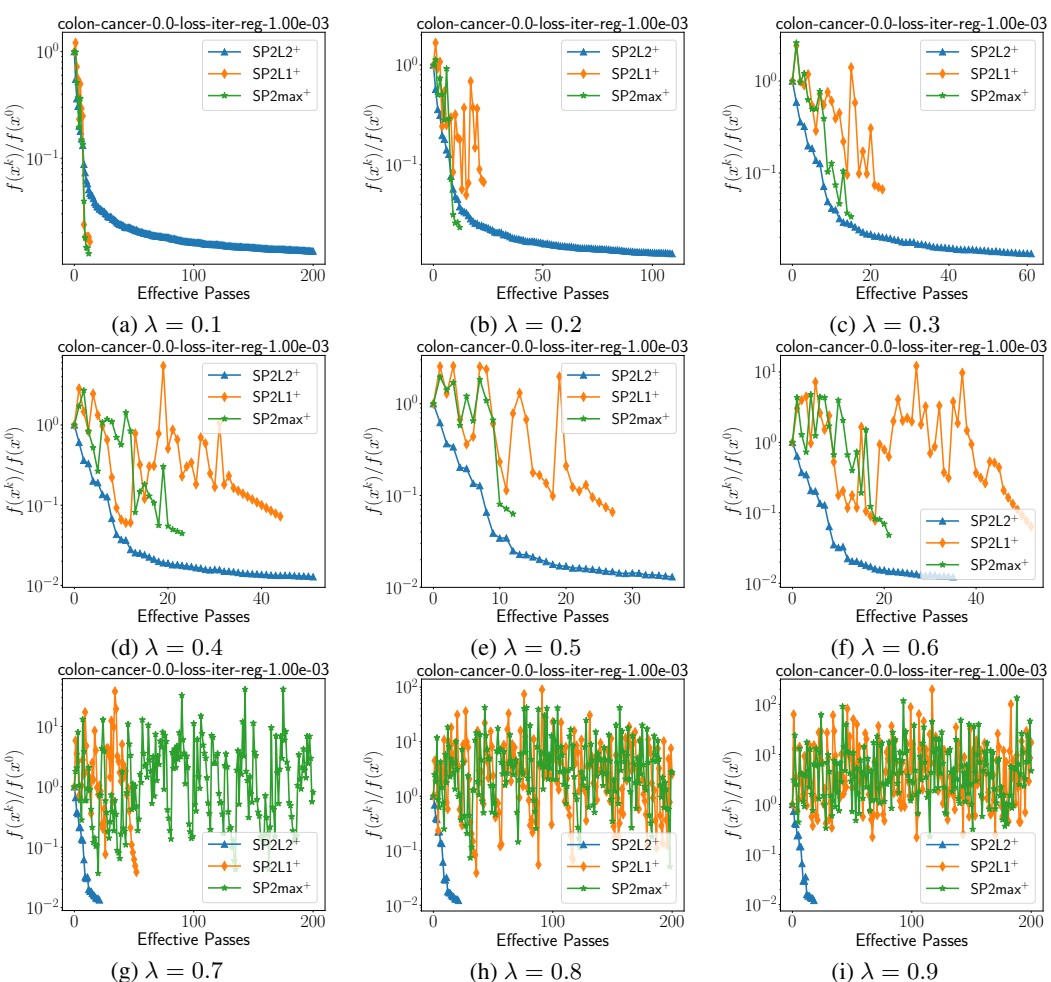

Figure 11: Colon-cancer: loss at each epoch with different $\lambda$.

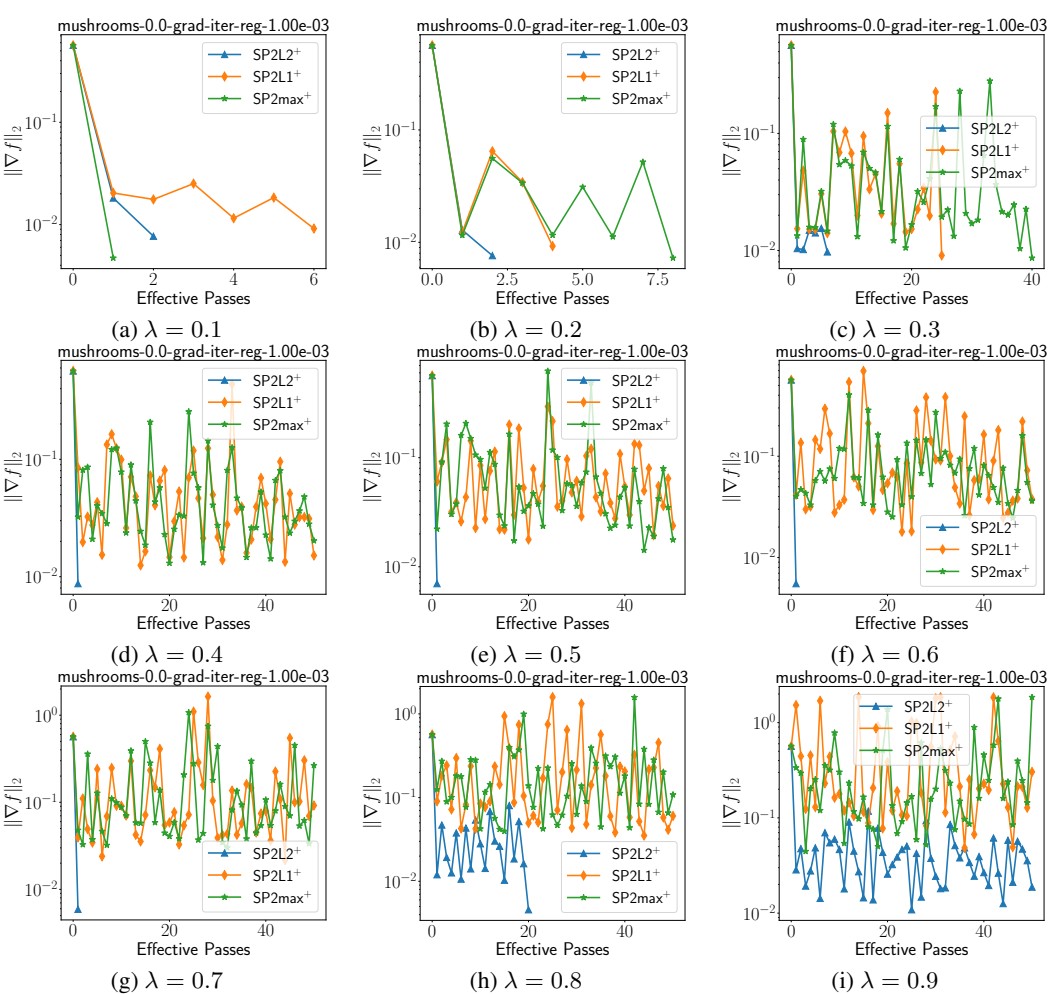

Figure 12: Mushrooms: gradient norm at each epoch with different $\lambda$.

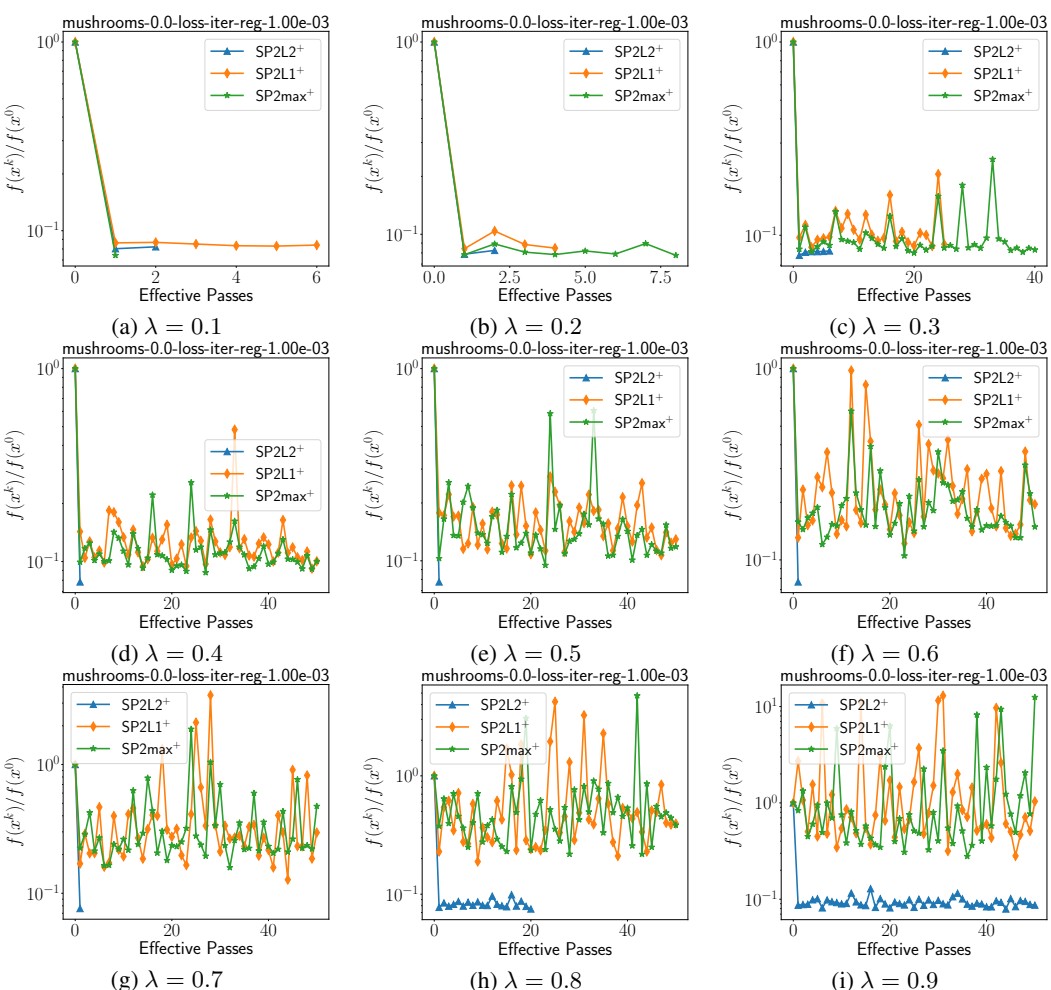

Figure 13: Mushrooms: loss at each epoch with different $\lambda$.

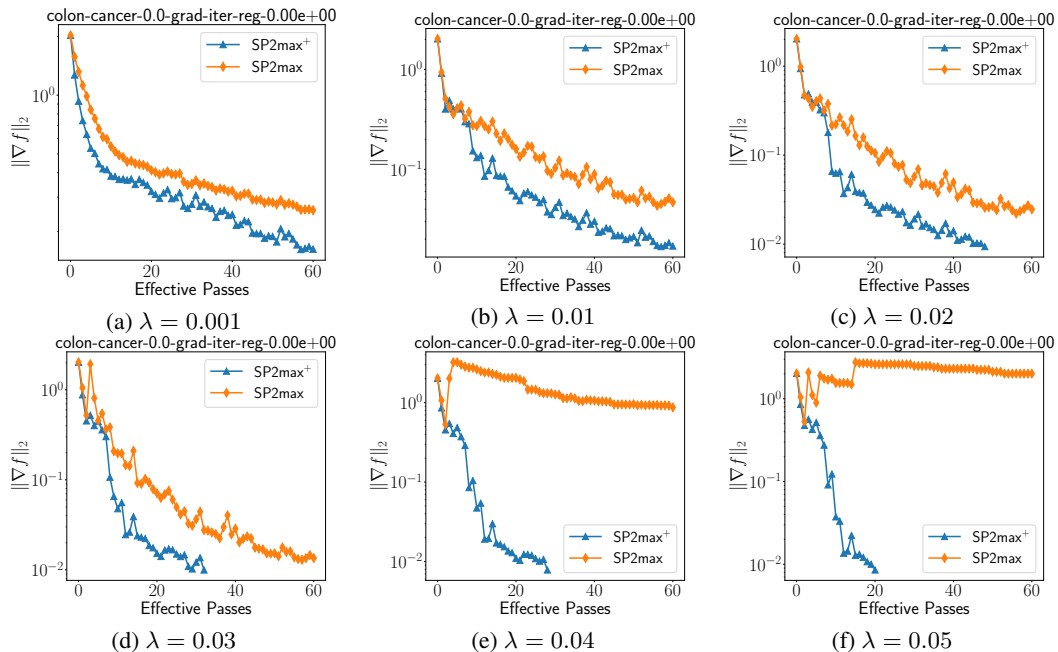

Figure 14: Colon-cancer: gradient norm at each epoch with different $\lambda$.

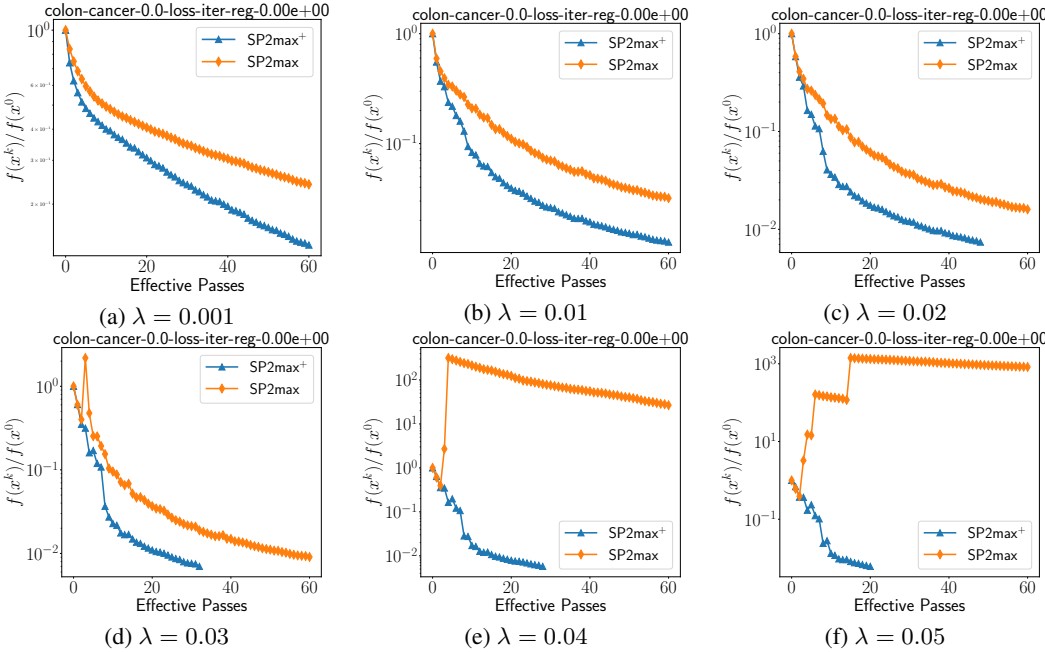

Figure 15: Colon-cancer: loss at each epoch with different $\lambda$.

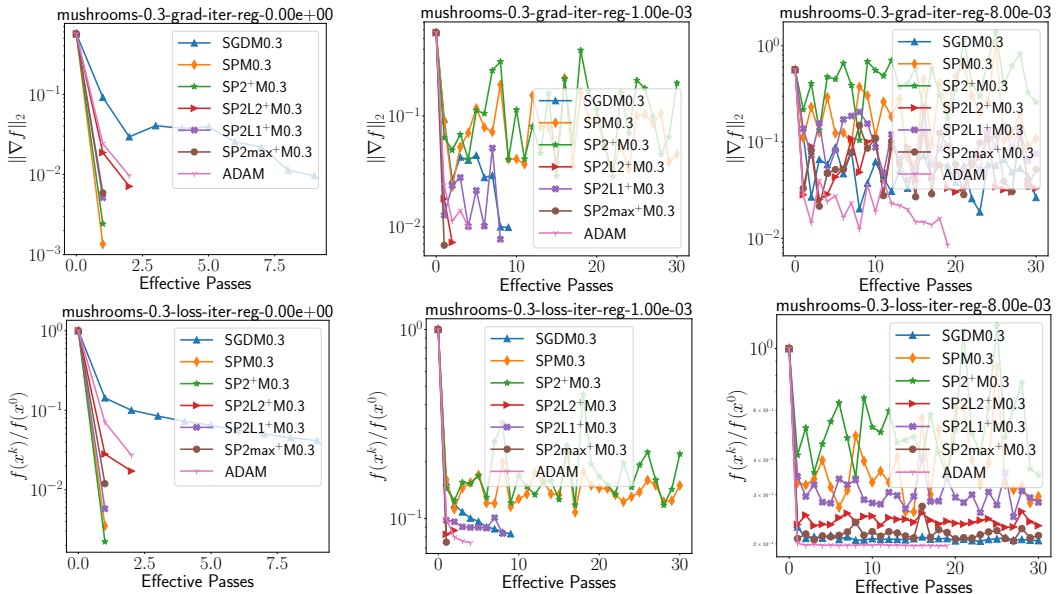

Figure 16: Mushrooms: gradient norm and loss at each epoch with momentum being 0.3.

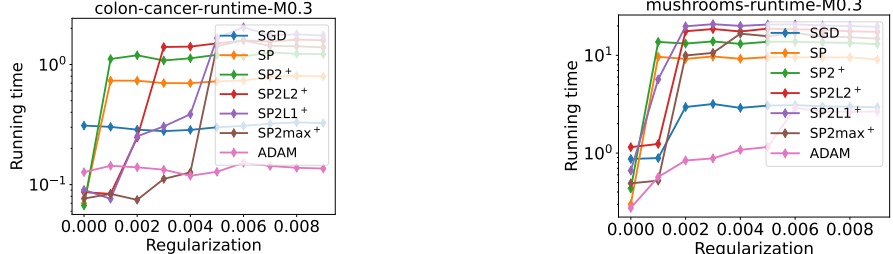

Figure 17: Running time in seconds.

