# OpenReview forum: "SP2 : A Second Order Stochastic Polyak Method"
_ICLR.cc/2023/Conference — ICLR 2023 poster_

### Official Review · Reviewer_fkL2 · 2022-10-13

**Confidence:** 5
**Correctness:** 4
**Technical Novelty And Significance:** 3
**Empirical Novelty And Significance:** 3
**Recommendation:** 6

**Clarity, Quality, Novelty And Reproducibility:**

The clarity and novelty are good.

There are some drawbacks in terms of the quality, as mentioned in "Weakness".


**Strength And Weaknesses:**


# Strength
The paper enjoys the following strength:

- The paper considers an interesting problem of solving non-linear
equations, which is potentially important and relevant to some
readers of ICLR.

- The paper is written clearly, and the proposed method consists of a
  simple idea.

- The paper covers several aspects of the proposed methods, including
  convergence theory, experiments, and extensions to the case of
  *quadratic with slack*.

- Full proofs are in the appendix in a single file, which makes the
  the paper easy to navigate.

# Weakness
The weakness of the paper manifests itself from several aspects:

- The assumption of Proposition 1 is too strong, which makes it not very
  interesting. First, it is a bit weird that the global minimizer
  $w^*$ has already shown up in the definition of $f_i$ (21), which
  is quite artificial. Second, $f_i$ is already quadratic,
  and local quadratic approximation does not make too much sense.
  Third, $H_i$ is positive semidefinite, which means that the
  quadratic constraint $f_i(w)=0$ is equivalent to the linear
  constraint that $w$ should lie in the null space of $H_i$ (if $H_i$
  is rank-deficient). This makes the convergence analysis trivial, as
  it is reduced to the situation that we already know (e.g., from the
  analysis of the Kaczmarz method). Finally, the discussion on the case
  where $H_i$ is invertible is misleading: In this case $H_i$ is
  positive-definite, the equation $f_i(w)=0$ has a unique
  solution, and one does not need to do a projection as solving
  $f_i(w)=0$ directly would suffice.
- Proposition 2 is a bit mysterious. Since it follows directly from
  Gower et al. (2021), that should be mentioned explicitly right after
  Proposition 2. Besides this, SP2+ itself is also mysterious and not
  very well motivated: In (18), intuitively, what is the benefit of
  considering a linearization of a quadratic approximation, instead of
  a linearization of the original function? That question also applies
  to Section 5.

- The two propositions stand in contrast to the name SP2 and very
  much also the purpose of the paper: Under the given assumptions the
  paper is essentially analyzing first-order methods (in a trivial
  way, based on prior works). The paper would be significantly
  strengthen if there is some solid analysis that reveals superlinear
  or quadratic rates of the proposed idea, as is typical for
  second-order methods.

- Section 5 would be a powerful technical contribution of the paper
  if it were more well-motivated. For example, what is the benefit of
  having two different slack formulations (L1 and L2)? What is the
  reason of resorting (30) for solving the L1 slack problem? After
  all, they seem to perform very similarly in the experiments.


- There are several sub-optimal aspects in experimental evaluation.
  First of all, the comparison to Newton's method is unfair, as it is well known
  that standard Newton's method is not always globally convergent. A slightly
  more considerate option is to compare with the damped Newton's
  method (as described in Boyd's book), which is guaranteed to
  converge, at least in the (strongly) convex case. Second, the
  plots should not be *function values versus epochs*, but it should
  be *function values versus running time*; the reason is that the
  methods under consideration have different time complexities at each
  iteration. Third, by just reading the paper, the reader has no idea
  how large the dataset in each experiment is. Since Sections 1
  and 2 make it an important point to design incremental methods that
  can work with mini-batches, it is important to run experiments on
  large datasets for which traditional offline methods run out of
  memory (e.g., Newton's method), otherwise, there is no reason to believe that online methods
  could perform better than offline methods. E.g., online methods
  would have very large variances and could easily fail in the case
  where some equation is far from being satisfied at the global minimizer.

**Summary Of The Paper:**

The paper considers the problem of solving finitely many non-linear
equations, under the assumption that a solution exists. The key idea
is projecting the current iterate onto the hyper-surface defined by
the quadratic approximation of the current equation of choice. The
first proposed method (SP2-GLM) considers the case where the non-linear equation is the
composition of some loss with a linear function. The second method
(SP2+) further linearizes the quadratic approximation. Some
convergence theory is derived for the two methods (Section 4). Extensions to the
case where there is an "approximate" solution are considered (Section
5). And experiments are demonstrated on several problems (Section 6).

**Summary Of The Review:**

I found the proposed method simple (a strength in my point of view), and some technical contributions are made (say in Section 3.1 and Section 5). But I also found Section 4 regarding convergence analysis and the experimental section unsatisfactory, as mentioned above in "Weakness".

At this moment, I made a borderline score, "marginally below the acceptance threshold".

Note that I am not an expert in this line of research. I will re-evaluate the paper based on the comments from other reviewers and the rebuttal.

---

> ### Author Response · Authors · 2022-11-11
> **10. The reader has no idea how large the dataset in each experiment is. Since Sections 1 and 2 make it an important point to design incremental methods that can work with mini-batches, it is important to run experiments on large datasets for which traditional offline methods run out of memory (e.g., Newton's method).**
>
> Answer: We will add some larger experiments and we will clarify the size of the dataset used in each experiment in the revised paper. In particular, the colon-cancer data set has 2000 features, making it already inappropriate to run full batch Newton, which would require solving  a 2000X2000 linear system.

---

> ### Author Response · Authors · 2022-11-11
> **9. The plots should not be function values versus epochs, but it should be function values versus running time; the reason is that the methods under consideration have different time complexities at each iteration.**
>
> We will include more plots with the x-axis is the running time. But please note we already have plots with running times in Figure 14. Furthermore, we find it problematic to change all the x-axises in the paper to time because time depends on the architecture, computing resources and implementation. As consequence reproducing time plots is virtually impossible.

---

> > ### Comment · Reviewer_fkL2 · 2022-11-11
> > **Running Time Report**
> >
> > I am not very convinced by this argument.
> >
> > *time depends on the architecture, computing resources, and implementation":
> >
> > There are already standard implementations for operations such as matrix-vector product. This should be the main computational cost for many of the methods. Use the same function to implement this matrix-vector product, and run the experiment on the same computer. I did not see why this is problematic.
> >
> > I am not suggesting switching all figures into running times. But a figure reproducing Figure 1 with running times would only make the paper more convincing (if SP2 is faster than ADAM or SGD), or at least make the paper trustworthy (if the running times are comparable or slower).
> >
> > Already two of three reviewers (66%) have this concern.

---

> > > ### Author Response · Authors · 2022-11-11
> > > **Will upload time results soon**
> > >
> > > Understood, we will reproduce Figure 1 but with running times and upload it soon. We will also try to include more time plots (time permitting).
> > >
> > > But again, we want to emphasize that even matrix-vector products depend on architecture. In particular, the difference in time for computing a matrix vector product on a GPU or CPU can be immense. This is even if one uses the same BLAS routines, let alone different algorithms for computing a matrix-vector product. In terms of implementations, in python there exists several alternatives to computing Hessian-vector products using tensorflow, pytorch, jax ... etc all of which have different time traces depending on dimension. This is why time plots are, at the very least, very hard to reproduce.

---

> ### Author Response · Authors · 2022-11-11
> **8. The comparison to Newton's method is unfair, as it is well known that standard Newton's method is not always globally convergent. A slightly more considerate option is to compare with the damped Newton's method (as described in Boyd's book), which is guaranteed to converge, at least in the (strongly) convex case.**
>
> Thank you for the comments. We will compare with the damped Newton's method, but in the case of these highly non-convex problems damped Newton also does not converge. Nor is there any agreed method for choosing the dampening parameter. Our only hope here would be to choose a dampening parameter large enough so that the resulting method is practically gradient descent. In fact, no second order method that we are aware of would converge on these highly non-convex examples. This is why SP2 is such a unique second order method. This was the purpose of these experiments, not to make a fair comparison to Newton type methods, but to highlight how unique SP2 is as a method.

---

> > ### Comment · Reviewer_fkL2 · 2022-11-11
> > **Other second order methods**
> >
> > Thanks for the clarification.
> >
> > I now have a new observation.
> >
> > Several stochastic second-order methods exist (as mentioned in the paper). It seems that the paper did not compare with them. What is the reason for that?

---

> > > ### Author Response · Authors · 2022-11-11
> > > **Second order and non-convexity**
> > >
> > > The reason we did not compare to other second order methods is because we were focused on non-convex learning problems. Which is why we emphasize non-convex GLMS in Lemma 2, the non-convex test functions in Section 6.1 and non-convex matrix completion in Section 6.2. The stochastic second order methods that we are away of rely on convexity in their design, and it would not make sense to apply them here. This again highlights how SP2 and SP2$^+$ are quite unique. Indeed, our follow up work is exploring the use of SP2$^+$ for training DNNs, which poses several additional challenges.
> > >
> > > But we do concede that we could have made it clearer that our target was a stochastic second order method that works in the non-convex setting. We will clarify this in the revision.

---

> ### Author Response · Authors · 2022-11-11
> **7. Section 5 would be a powerful technical contribution of the paper if it were more well-motivated. For example, what is the benefit of having two different slack formulations (L1 and L2)? What is the reason for resorting (30) for solving the L1 slack problem? After all, they seem to perform very similarly in the experiments.**
>
> We will add  further motivation of Section 5 in the revised paper.  The linear and quadratic parameterizations of the slack term are both natural, so we felt that it made sense to compare them experimentally.  As you mentioned, they do turn out to behave similarly in practice however this was not clear to us a priori. Finally, we apologize but we did not understand the reviewer's comment regarding “resorting (30)”. Could the reviewer please explain and engage with us here?

---

> > ### Comment · Reviewer_fkL2 · 2022-11-11
> > **(30)**
> >
> > I meant that, given this L1 slack formulation, why do we resort to solving (30)?
> >
> > Never mind. I re-read the paper and now I find it clear.

---

> ### Author Response · Authors · 2022-11-11
> **6. The two propositions stand in contrast to the name SP2 and very much also the purpose of the paper: Under the given assumptions the paper is essentially analyzing first-order methods (in a trivial way, based on prior works). The paper would be significantly strengthen if there is some solid analysis that reveals superlinear or quadratic rates of the proposed idea, as is typical for second-order methods.**
>
>  We agree that this would be very desirable, but also very challenging to achieve for stochastic second order methods. And again we respectfully disagree that arriving at Proposition 1 was trivial. It was hard work to derive a simple analysis in this setting. As we pointed out earlier, it is indeed a generalization of the analysis of Kaczmarz, but it does not result from applying the analysis of Kaczmarz. Finally, though we do not have super linear results, the convergence rate $\rho$ in Proposition 1 is unlike any first order method, since it does not depend on the condition of the $H_i$ matrices. Indeed, the spectrum of $H_iH_i^+$ is always contained in $\{0, 1\}$. This is one of the main advantages of second order methods: To be independent of the conditioning of the problem. To further emphasize how this is also not an application of Kaczmarz, note that the Kaczmarz method applied to this problem would have a rate of convergence of
> $1- \lambda_{\min} (A^TA)/||A||_F^2$.
>
> where $A = \sum_{i=1}^n H_i$ which does depend on the square of the scaled condition number. As such Kaczmarz is very sensitive to the conditioning of the matrix $A$.

---

> ### Author Response · Authors · 2022-11-11
> **5. Proposition 2 is a bit mysterious. Since it follows directly from Gower et al. (2021), that should be mentioned explicitly right after Proposition 2. Besides this, SP2+ itself is also mysterious and not very well motivated: In (18), intuitively, what is the benefit of considering a linearization of a quadratic approximation, instead of a linearization of the original function? That question also applies to Section 5.**
>
> Thank you for the comments. (1) We will mention Gower et al. (2021) right after Proposition 2 in the revised paper. (2) The benefit of linearizing the quadratic, or rather, taking steps towards solving the local quadratic, as opposed to linearizing the original function, is ultimately to arrive at a method that is more stable and converges faster. Using only the local linearization results in the SP method which can be unstable and thus requires more tuning to work well (see for example SPM = SP with momentum in the middle of Figure 4). By instead expending a few linearization steps on the local quadratic, the hope is that since the quadratic is a better model, the resulting method will require less tuning, as is often observed in second order methods. This is something we can even observe in Figures 4 and 5 (by a slight margin), that SP2$^+$ has an improved convergence as compared to SP.

---

> ### Author Response · Authors · 2022-11-11
> **4. Finally, the discussion on the case where Hi is invertible is misleading: In this case Hi is positive-definite, the equation fi(w)=0 has a unique solution, and one does not need to do a projection as solving fi(w)=0 directly would suffice.**
>
> We apologize if our comments about this specific case where one $H_i$ is invertible is misleadings. The fact that $f_i(w)=0$ has one solution is exactly why SP2 will converge after sampling this $i$ index. We will gladly clarify this in the revision.

---

> > ### Comment · Reviewer_fkL2 · 2022-11-11
> > **Should Hi be invertible?**
> >
> > As a rule, I think in this line of research (e.g., on methods that alternate between projecting onto the space defined by some equations), a basic assumption should be that, at each iteration, the chosen equations should not uniquely define the solution (otherwise we arrive at a trivial case where projection is not needed and we can just solve the equations).
> >
> > For this reason, I think the discussion of the case where Hi is invertible is not very interesting. Any reasonable algorithm should directly reach the global minimizer if Hi is invertible.
> >
> > The issue then seems to be that there is something wrong with Karzmarz, or the way how Karzmarz is compared.

---

> > > ### Author Response · Authors · 2022-11-11
> > > **Comment retracted, did not understand Karzmarz comment**
> > >
> > > Understood, we will retract our comment about this special case on one $H_i$ being invertible. We will upload a revised version of the paper in the following days for you to verify.
> > >
> > > We did not, however, understand your comment about Kaczmarz. Did you mean SP2$^+$ ? Could you please clarify?

---

> ### Author Response · Authors · 2022-11-11
> **3. Hi is positive semidefinite, which means that the quadratic constraint fi(w)=0 is equivalent to the linear constraint that w should lie in the null space of Hi (if Hi is rank-deficient). This makes the convergence analysis trivial, as it is reduced to the situation that we already know (e.g., from the analysis of the Kaczmarz method).**
>
> Answer: We have carefully thought about the reviewers comment here, and indeed find that there is a connection to the analysis of Kaczmarz. But we respectfully disagree that it is trivial and we disagree that this reduces to the analysis of Kaczmarz. In fact, quite the opposite, our analysis is in fact a generalization of the analysis of Kaczmarz. We find that this is a very interesting insight, one that even opens further questions, as we detail now. To clarify this, indeed the constraint $f_i(w) =0$ now reduces to $H_i(w -w^*) =0$. Kaczmarz is a method for solving linear systems, but it does so by sampling the rows. That is, Kaczmarz solves a linear system Ax =b by at each iteration projecting onto $e_i^T Ax = e_i^T b$ where $e_i$ is a unit coordinate vector. Because of this, Kaczmarz is often grouped together with projection based algorithms that solve at each iteration $P_i Ax = P_i b$, where $P_i$ is a projection. At face value, this is different from our setting where we must solve $ \sum_{i=1}^n H_i (w-w^*) =0$ by sampling the matrices $H_i$, and not the rows of $\sum_{i=1}^n H_i$ or any projection of this matrix. But in terms of analysis, our analysis is in fact a direct extension of the Kaczmarz analysis that handles *any* decomposition of the matrix $A$ into a finite sum. To see this, first note that $A = \sum e_ie_i^TA$. Consequently we can see the Kaczmarz method as sampling from the sum $e_ie_i^TA$ and solving at each iteration $e_ie_i^TA x= e_ie_i^Tb$. Thus, in the setting of minimizing sums-of-quadratics, our method is an extension of Kaczmarz to handle any such decomposition into sums (not necessarily the row decomposition), and our analysis extends the analysis of Kaczmarz to handle any decomposition. We find that this is a very keen observation, one that we had not noticed (despite being familiar with Kaczmarz and its analysis). This even opens up questions as to how to design such decompositions to arrive at new variants of the Kaczmarz method. We will clarify this in the appendix. In any case, we do not think this trivial, nor does this detract in any way from the insight this analysis gives or the value of our contribution. Again, quite the opposite, we find the connection to Kaczmarz (specifically a generalization of that analysis) only adds to the value of our contribution. Finally, though we refined our analysis to be as simple as possible, we disagree that it is trivial. We have worked hard to make it simple and clear.

---

> > ### Comment · Reviewer_fkL2 · 2022-11-11
> > **Prop 1 and Karzmarz**
> >
> > Thanks for the detailed clarification.
> >
> > In Eq. (89), the problem is reduced in a (mini-batch) Karzmarz form: project the current iterate onto some affine subspace defined by the selected linear equations.
> >
> > Then we have two choices:
> > - we can still follow the proof of Prop 1, and obtain a convergence rate, say rate 1
> > - we can invoke existing results for the Karzmarz method (can we? I am not sure), then we can obtain a rate, say rate 2.
> >
> > I would recommend comparing the two rates.

---

> > > ### Author Response · Authors · 2022-11-11
> > > **We cannot invoke Karzmarz rates here**
> > >
> > > Thanks for the follow up and for engaging.
> > >
> > > From (89) we cannot directly invoke any theory of Karzmarz or any block variant of the Karzmarz method. The reason being, that SP2 when applied to sums of quadratics is not an instance of the Karzmarz or block Kaczmarz method. Rather, the opposite holds: we can re-write the proof of convergence of Kaczmarz as a special case of our analysis where specifically $H_i = e_ie_i^T A$ and $b_i = H_iw^*$ as we explained before. Said in another way, we could apply our proof technique to analyse the Kaczmarz method. The result would be the well known linear rate of Kaczmarz that depends on the scaled condition number. But the opposite, applying the Kaczmarz analysis to our case  is not possible.
> > >
> > > To further emphasize, there is no notion of mini-batch Kaczmarz. The Kaczmarz methods subsamples rows (not matrices). The block Kaczmarz subsamples blocks of rows, but not sums of matrices which is what we need here.
> > >
> > > Finally, if you compare the proof in Strohmer & Vershyni 2007 to our proof of Proposition 1, their similarity is not even apparent. But, as highlighted in our previous comment, our proof generalizes the Kaczmarz proof technique.
> > >
> > > Please let us know if this answers your question, and if not, we can continue to discus.

---

> ### Author Response · Authors · 2022-11-11
> **2. fi is already quadratic, and local quadratic approximation does not make too much sense.**
>
> Answer: Indeed but please note that we are not suggesting this as a method for minimizing quadratics, but instead, analyzing our method on a quadratic to get some insight into its convergence. It also gives insight into how our methods would converge locally around a minimizer where the local quadratic approximation suffices. But note that this is not about suggesting an appropriate method for minimizing quadratics.

---

> ### Author Response · Authors · 2022-11-11
> **1. It is a bit weird that the global minimizer w∗ has already shown up in the definition of fi(21), which is quite artificial.**
>
>  Answer: Thank you pointing this out, in fact there was a small mistake here that made it confusing. We meant to say that $w^*$ is a constant vector in $\mathbb{R}^d$. We then define the loss functions $f_i$ in (21), for which we can show that $w^*$ is a global minimizer. Does this clarify the issue? If not, please engage with us here.

---

> ### Author Response · Authors · 2022-11-11
> **General Comments**
>
> Thank you for your time and extensive review. We have given your review careful consideration and addressed all the points you raised below. We would like to single out two of these points: a) The analysis in Proposition 1 is trivial. Here we believe the reviewer is conflating the hard work of a simple and clean analysis with it being trivial. Please note we have recognized the reviewers point on a connection to the analysis of Kaczmarz (see below for details), but we find that this is not a trivial connection but rather a generalization of the analysis of Kaczmarz.  Thus we kindly ask the reviewer to reconsider this point, or engage with us now during the discussion.
>
> The other point b) is a lack of “superlinear or quadratic” convergence. This would be considered the holy grail for a stochastic second order method, and asking for such a contribution in addition to our contributions is not reasonable. Indeed, we are not aware of any stochastic second order method that is incremental (this excludes large batch size subsampled Newton in high probability) that achieves a superlinear or quadratic convergence. Thus we politely ask that the reviewer instead consider what are our claims and corresponding contributions.

---

> > ### Comment · Reviewer_fkL2 · 2022-11-11
> > **Thanks for the rebuttal**
> >
> > Dear authors,
> >
> > Thanks for the rebuttal. It is very clear and I greatly appreciate it.
> >
> > I added some comments regarding point (a) below.
> >
> > For point (b), I would not require such proof if it is really something tough.
> >
> > Just out of curiosity (or the authors could actually discuss it in the paper): What is the challenge or difficulty of proving a superlinear rate for stochastic second-order methods?
> >
> > Minor points: There are some missing references in the appendix, annotated as **[cite]**.
> >
> >
> > Best,
> > Reviewer kkL2

---

> > > ### Author Response · Authors · 2022-11-11
> > > **Thank you for engaging, difficulty with superlinear**
> > >
> > > Dear Reviewer,
> > >
> > > thank you for engaging, and your follow up remarks. We will answer you follow up below each itemized comment.
> > >
> > > As for the difficulty of superlinear convergence, there are several challenges. First, in the deterministic setting, superlinear convergence is already rare. We are aware of only two settings were it occurs 1. Quadratic local convergence of Newton's method and 2. Local superlinear convergence of quasi-Newtons.
> > >
> > > The proof for 2. is so challenging that from the moment the first asymptotic proof of superlinear convergence appeared (M. Powell 1971)  it has taken 50 years to develop an explicit superlinear convergence rate (Anton Rodomanov & Yurii Nesterov 2022). Even an explicit linear convergence was missing (Kovalev et al 2020) until only 2020. So I will leave the difficulty of analysing quasi-Newton aside.
> > >
> > > As for local quadratic convergence of Newton's method, it only appears if we exactly solve the Newton system, and only if we are close to the solution. For stochastic methods, it is generally not possible to even monitor when we are close to the solution. Furthermore, we can only admit approximate solutions to the Newton system, since we have only stochastic gradients and Hessian matrices. The only workaround for stochastic methods that we are aware of is to keep in memory and estimate of every Hessian of every sampled function, so as form a full Newton system (Anton Rodomanov & Dmitry Kropotov 2015), but this quickly becomes computationally infeasible.  In any case, this only holds for strongly convex objectives (or a similar notion of strong self-concordancy) which is the setting we are interested here.

---

> > > > ### Comment · Reviewer_fkL2 · 2022-11-11
> > > > **Summary of My Review and Recommendation**
> > > >
> > > > Dear authors,
> > > >
> > > > Thanks for the reply. It is well received. It has now addressed all of my concerns. I also learned a lot from the rebuttal.
> > > >
> > > > I increased my score and confidence by 1.
> > > >
> > > > @AreaChair: I would be happy if the paper is accepted after some minor revisions on clarity (as described in the rebuttal), eg, add some discussions of the proposed method vs. second-order methods, vs. Karzmarz methods.
> > > >
> > > > Best,
> > > > Reviewer fkL2

---

> > > > > ### Author Response · Authors · 2022-11-12
> > > > > **Thank, revision**
> > > > >
> > > > > Dear reviewer,
> > > > >
> > > > > Thank you for your increased score and confidence. Moreover, thank you for taking the time to engage with us here.
> > > > >
> > > > > Early next week, hopefully by Tuesday, we will also upload the revised paper for your consideration.
> > > > >
> > > > > Regards,
> > > > > The authors

---

> ### Author Response · Authors · 2022-11-16
> **Revision, time plots and block Kaczmarz**
>
> Dear Reviewer,
>
> We have just now uploaded our revised paper. In particular we have included five more figures with time plots, where we found that SP2 was still very competitive in terms of time taken. Even more so when we increased the dimension of the non-convex experiments in Figures 3 and 9.
>
> Furthermore, we reconsidered the proof of Proposition 1 in connection to the proof of block Kaczmarz. And we now realize that you were right in your initial remark. Namely, there is a way of applying the analysis of the block Kaczmarz method to arrive at our rate of convergence, see the footnote on page 5 and Remark 1 in the appendix. We have thanked you for pointing this out.
>
> If there is anything else you would like to see in the revision, please reach out to us here.
>
> Kind regards,
> The Authors.

---

### Official Review · Reviewer_DF7u · 2022-10-24

**Confidence:** 4
**Correctness:** 3
**Technical Novelty And Significance:** 3
**Empirical Novelty And Significance:** 2
**Recommendation:** 6

**Clarity, Quality, Novelty And Reproducibility:**

The idea of using stochastic Polyak step size and second-order approximation is not novel, but the authors did combine them relatively well.

**Strength And Weaknesses:**

The strengths of this paper come from the nice framework to approximate the second-order function. It also proposes various different ways to solve the quadratic problem, including discussions using a generalization of the interpolation property.

There are potentially several limitations of this method. Firstly, it is not clear whether the quadratic subproblem can be solved efficiently in a big-data machine learning setting. In addition, most proposed methods only can approximate the solution of those problems, which poses some difficulty in analyzing these method theoretically and practically. Secondly, since the convergence theory is for quadratic functions, it is unclear if this method would have theoretical guarantee in most of the ML setting with interpolation (which is the motivation for the problem). Finally, the experiment and theoretical setting does not seem to match each other.

Other comment: The detailed algorithms should be stated separately in a box to avoid any confusion for the readers.

**Summary Of The Paper:**

This paper consider the interpolation settings. It proposes to use a similar approach as the stochastic Polyak step size (solving the interpolation equations). However, instead of using the first-order approximation, this paper uses the local second-order information of the model. The authors then listed several approaches for this, namely SP2 (closed form solution applying for generalized linear models), SP2+ with approximating the solution to subproblem, and other versions with slack formulation.

**Summary Of The Review:**

This paper has some good contributions, though the practicality of this method is in question. I would like to hear from the authors regarding the limitations of this papers.

---

> ### Author Response · Authors · 2022-11-11
> **5. The detailed algorithms should be stated separately in a box to avoid any confusion for the readers.**
>
> This is a good suggestion. We will state the algorithms separately in a box in the revised paper. Thanks.

---

> ### Author Response · Authors · 2022-11-11
> **4.  The experiment and theoretical setting does not seem to match each other.**
>
> Dear reviewer, could you please clarify the sense in which they do not match? We will happily answer in the discussion and update our experiments accordingly.

---

> > ### Author Response · Authors · 2022-11-16
> > **Please detail concern**
> >
> > Dear reviewer, we would very much like to address your concern here, but we are still unsure as to what is the mismatch you refer to. Could you please specify to give us the opportunity to address your concern before the discussion deadline ends in 2 two days? Furthermore, we have uploaded a revised version of the paper with five new plots. Perhaps you concern has been addressed in the revision? We look forward to hearing back from you. Thank you for your time, the Authors.

---

> > > ### Comment · Reviewer_DF7u · 2022-11-17
> > > **Thanks for your reply**
> > >
> > > Dear authors,
> > >
> > > Thank you for your revision and sorry for the late reply. The mismatch I mentioned is that the theory of your method mostly contributes to quadratic functions, while the experiment setting is nonconvex. I will look at the revision later and update my review, in case you have already included/ discussed the possibility of having theoretical results for more general settings.
> > >
> > > Thanks.

---

> > > > ### Author Response · Authors · 2022-11-17
> > > > **Convergence theory for non-convex is hard, additional numerics**
> > > >
> > > > Dear reviewer, thank you for engaging. Ok, understood. But we would like to emphasize that our convergence theory in Proposition 1 and 2 are rather more of a sanity check. Indeed, we are not interested in developing a method for minimizing sum of quadratics. Instead, our objective was develop a stochastic 2nd order method (SP2) that can be applied to non-convex objectives, such as the highly non-convex problems in section 6.1 and the matrix completion problem in section 6.2. As such, in our revision we have focused on including more numeric results to emphasize this.
> > > >
> > > > We agree that in this sense, there is a mismatch. But, in terms of additional theoretical results, the test functions in section 6.1 are non-convex, non-smooth and non-Lipschitz. There is no convergence theory for such functions for any method. Indeed, finding a stationary point of non-convex,  non-smooth and non-Lipschitz is NP-hard. On the other hand, it might be possible to produce a convergence theory for the matrix completion problems in section 6.2. Though matrix completion is non-convex, this a very specific non-convex problem for which there do exist meaningful convergence guarantees. But now, with only one more day left in the discussion phase, we do not have enough time to produce, and carefully check, an addition convergence theory for matrix completion. We hope instead the reviewer can instead appreciate the additional numeric results we have included.

---

> ### Author Response · Authors · 2022-11-11
> **3. Since the convergence theory is for quadratic functions, it is unclear if this method would have theoretical guarantee in most of the ML setting with interpolation**
>
> We could extend the analysis on sums of quadratics to a local analysis for smooth and convex functions, where the second order Taylor approximation is accurate. This would relatively standard. But in general, it is very challenging to derive meaningful theoretical guarantees for stochastic second order methods.

---

> ### Author Response · Authors · 2022-11-11
> **2. Most proposed methods only can approximate the solution of those problems, which poses some difficulty in analyzing these method theoretically and practically**
>
>  We agree, the approximation we use around SP2$^+$, and its multistep version in Section C.4, make it harder to analyze. But still, we do offer insight here, in particular aside from Proposition 2, we also show in Section C.4 that the multistep version of SP2$^+$ does converge to SP2. This in turn means we can perfectly control the quality of this approximation. But still, it is extremely challenging to arrive at a meaningful analysis of these, and other, stochastic second order methods.

---

> ### Author Response · Authors · 2022-11-11
> **1. It is not clear whether the quadratic subproblem can be solved efficiently in a big-data machine learning setting.**
>
> Our approach using steps of Newton-Raphson, such as SP2$^+$, can scale to any dimension. Indeed, this follows because SP2$^+$ uses only Hessian-vector products that can be computed efficiently with backpropagation and cost, at most, as much as 5 times the cost of a gradient. Thus despite being a second order method, SP2$^+$ has an iteration cost and memory footprint that is linear in the dimension d. Therefore there are virtually no limitations in dimension, and these methods can be applied to big data problems, making SP2$^+$ a very unique second order method

---

> ### Author Response · Authors · 2022-11-11
> **General Comment**
>
> Dear Reviewer, thank you for your time and thoughtful remarks on our paper. Please find below a point by point analysis of each of your questions. We look forward to engaging with you further during the discussion phase.

---

### Official Review · Reviewer_etGz · 2022-10-26

**Confidence:** 4
**Correctness:** 3
**Technical Novelty And Significance:** 2
**Empirical Novelty And Significance:** Not applicable
**Recommendation:** 5

**Clarity, Quality, Novelty And Reproducibility:**

Novelty: The main novelty of this paper is extending the linearized constrained into quadratic constraints and proposing closed-form solutions for various losses.

**Strength And Weaknesses:**

Empirically their proposed methods shows outperform other existing optimization methods in the literature.
Weakness: their theoretical analysis holds just for too limiting settings.

**Summary Of The Paper:**

SP (Stochastic Polyak step size) can be interpreted as a method specialized to interpolated models since it solves the interpolation equations. SP can be interpreted as a projection into a stochastic-constrained linearization of the objective function. The main idea of this paper is that it extends this constrained linearization to a quadratic approximation of the objective (SP2). They show a closed-form solution for this minimization of some objectives common in machine learning such as quadratic loss. In general, there may not be a solution for SP2 minimization, so they introduce SP2+ that contains two steps: 1- minimizing on a linearized objective and 2- minimizing on a quadratic approximation of the objective. They provide a convergence guarantee for SP2(+) for a quadratic loss in an interpolation setting.


**Summary Of The Review:**

Comments:
1- The empirical results are based on the number of epochs used through training. However, the iteration costs for SP2 and SP2+ are greater than SP or SGD. Therefore the right criteria would be wall clock time. So I suggest to change all x-axis changes to the wall-clock time so a fair comparison is doable.

2- In your theoretical result, specifically Prop 1 and 2, if we assume d > n, and H_i is non-zero only in the (i,i) element. This setting satisfies your assumptions for Prop1 and 2. For this setting, \rho would be 1 and your analysis won’t show a convergence.

3- Since your SP2 method needs a Matrix-vector product, it would be useful to show that for a simple loss how some can implement it efficiently.

4- It will be nice to first give the original formula of SP method before eq 4.

---

> ### Author Response · Authors · 2022-11-11
> **4. It will be nice to first give the original formula of SP method before eq 4.**
>
> Thank you for the comments. We will add the original formula of SP method before eq(4) in the revised paper.

---

> ### Author Response · Authors · 2022-11-11
> **3. Since your SP2 method needs a Matrix-vector product, it would be useful to show that for a simple loss how some can implement it efficiently.**
>
> Our methods do not require any explicit Matrix-vector products. The Hessian-vector products are computed by running a version of backpropagation, and thus cost the same as computing a gradient. For instance in pytorch one need only call torch.autograd.functional.hvp to compute a Hessian vector product. We will clarify this point in the revision.

---

> ### Author Response · Authors · 2022-11-11
> **2. In your theoretical result, specifically Prop 1 and 2, if we assume d > n, and H_i is non-zero only in the (i,i) element. This setting satisfies your assumptions for Prop1 and 2. For this setting, \rho would be 1 and your analysis won’t show a convergence.**
>
> This is not entirely correct. If $H_{ii}$ is non-zero only in the $(i,i)$-th element for every $i$, then we have that $H_i H_i^+ = e_ie_i^T$, where $e_i$ is the unit coordinate vector. In this case the matrix that defines $\rho$ in Proposition 1 is given by
> $$I -(1/n) \sum_{i=1}^n H_iH_i^+ = I - 1/n \sum_{i=1}^n e_ie_i^T =  (1-1/n)I \in \mathbb{R}^{d\times d}$$
> Thus $\rho = (1-1/n).$
> In the case of Proposition 2, it is correct that $\rho =1$. This is perhaps not so surprising, since the $f_i$ functions are not strongly convex, so we should not expect a linear convergence. It is in fact remarkable that due to Proposition 1 that SP2 still converges linearly in the absence of strong convexity. With regards to Proposition 2, we will add this comment to the paper.

---

> > ### Comment · Reviewer_etGz · 2022-11-23
> > **Convex case.**
> >
> > In the prop 1, you assume $H_i \geq 0$ and this means each $f_i$ is convex. In the interpolation setting when $d >>n$ we have that $f$ is convex and not strongly convex. Then your proposition doesn't show the convergence or it breaks the lower-bound for convex+interpolation setting.

---

> > > ### Author Response · Authors · 2022-11-23
> > > **Which lower bounds?**
> > >
> > > Dear reviewer, are you referring to the fact that SP2 converges linearly despite the absence of strong convexity? Indeed for convex + interpolation SGD converges at a rate of O(1/t) and does not enjoy a linear rate. But SP2 does not satisfy the same oracle as SGD since it requires access to the Hessian (the H_i matrices). Thus we don't see how this results breaks any lower bounds we are aware. Could the reviewer please clarify which lower bound for convex+interpolation are you referring to? Also, do you agree with our previous answer that in Proposition 1 we have that $\rho = 1-1/n$ for the example given?
> > >
> > >
> > > Finally, please note that for us the interpolation setting refers to Assumption 1 holding, which does not require $d>>n$. Instead, it requires that the loss on all samples be zero. This can happen even when $d< n$.

---

> ### Author Response · Authors · 2022-11-11
> **1. The empirical results are based on the number of epochs used through training. However, the iteration costs for SP2 and SP2+ are greater than SP or SGD. Therefore the right criteria would be wall clock time. So I suggest to change all x-axis changes to the wall-clock time**
>
> Thank you for the comments. Please note that we do provide the running time comparison for the logistic regression problems in Figure 14 in the appendix. We will also include several more plots comparing running time, which we agree is important. But, we find that changing all of the x-axes in the paper to time is problematic, since time depends on the architecture, computing resources and implementation. This can make time results hard to interpret and impossible to reproduce. Would the reviewer agree to having simply more plots based on time?

---

> ### Author Response · Authors · 2022-11-11
> **General comment**
>
> Dear Reviewer, thank you for your questions and comments. We have addressed all of them below. In particular, please note that we have completely addressed your issues 2., 3. and 4. As for issue 1. We have already included time plots, and will include more. As such, could the reviewer kindly review their score? In particular, we noticed you have marked that in terms of correctness “Some of the paper’s claims have minor issues”. Was this because of the supposed issue around the convergence rate $\rho$? Please note we have clarified that this is not an issue, so we kindly ask that the reviewer re-evaluate their position on correctness. If this is still not clear, we only ask that you please engage with us now during the discussion phase to clarify this point. Yours truly, the authors.

---

> ### Author Response · Authors · 2022-11-16
> **Revision and time plots**
>
> We have just now uploaded our revised paper, where we correct all of the issues you raised. In particular we have now include five more figures with time plots. Would you mind checking the revision and letting us know if this addresses your concerns? And if so, would you change your correctness/score accordingly?

---

### Author Response · Authors · 2022-11-16
**Revision uploaded**


Dear Reviewers and AC, we have just uploaded a revised version of our paper, where we address the questions and concerns of the reviewers. Our new changes are in red.

In particular, we have added several new experiments comparing our new methods SP2 and SP2^+ to ADAM, SP, SGD in terms of time taken, see the updated Figures 1, 2, 3, 8 and 9. Though as to be expected SP2 was less competitive in terms of time (as opposed to epochs), we found that our main conclusion holds, that is, that SP2 is very competitive for solving non-convex objectives, moreover, it is perhaps one of the only second order method capable of efficiently solving non-convex programs. Furthermore, we have also added larger dimension non-convex problems in Figures 3 and 9, which are typically unsolvable. Remarkably, SP2 is able to find a global solution, and together with SP are to the most efficient methods when considering time taken. In terms of epochs, SP2 is by far the most efficient method, see Figure 9. Finally, we have given an unfair advantage to both SGD and ADAM in these experiments, since we used the first 100 epochs to tune the step size of these methods over a grid search of
[0.0001, 0.0005, 0.001, 0.005, 0.01, 0.05, 0.1].
For our methods SP2, SP2^+ and for SP we tuned the stepsize over the grid
[0.005, 0.01, 0.05, 0.01].
 This makes our results even more noteworthy.

Since adding time comparisons was one of the main issues raised by reviewers etGz and  fkL2, we hope that this revised version of our papers addresses their concerns.

Kind regards, the Authors

---

> ### Comment · Reviewer_fkL2 · 2022-11-16
> **Revision Received**
>
> Dear authors,
>
> Thanks for the revision. I have just read it.
>
> **@AreaChair**: I think, after significant efforts on revision, this is now a solid paper with several strengths: (1) detailed discussion on related works, (2) well-written, (3) good performance with many figures, (4) novel algorithms (SP2 and SP2+), (5) there are multiple potential applications of the proposed methods (I might even contact the authors for further discussions on this after the review period).
>
> I think this paper deserves publication at ICLR 2023. I would like to give a score of 7 if there were such an option. (Perhaps I am a strict reviewer. I have never given a score of 7/8 or higher in my limited review experience; I think papers with scores >=8 should already be very good at the time of submission, not after revision)
>
> I would also like to have discussions with the other reviewers/area chairs to reach a consensus.
>
> Best,
> Reviewer fkL2

---

> > ### Author Response · Authors · 2022-11-16
> > **Thank for your time**
> >
> > Thank you for the positive comments and thank your for the time and work you have dedicated as reviewer. Just to note, the score range for ICLR is from 1 to 10. In particular the score 6 (weak acceptance) is below the average over accepted papers for ICLR 2022 which was  a score of 6.63. A score 7 is just above the average acceptance score for 2022, and below that of a spotlight paper (score 7.32 on average). These statistics on ICLR 2022 can be found here guoqiangwei.xyz/iclr2022_stats/iclr2022_statistics.html

---

> > ### Comment · Area_Chair_MPxq · 2022-11-22
> > **Adjusting score**
> >
> > Dear reviewer fkL2,
> >
> > I think you can adjust the score using the "edit" option in the open review. You can look at the upper-right corner of your review, and you will find the button that looks like a pen on the pad. Click that button and then you will be able to edit your score.
> >
> > Best,
> > AC

---

### Author Response · Authors · 2022-11-18
**Minor revision uploaded, thank you for your time**

Dear AC and Reviewers,

We have just now uploaded another minor revision to our paper. The main highlight in this revision is that we have improved the style of plots. We were particularly happy with our new results on the non-convex experiments, and thus felt they deserved to be as clear as possible.

The discussion period is now ending, and we thank reviewers fkL2 and DF7u for interacting. We also believe we have addressed all four concerns of reviewer etGz, including their main concern in adding more time plots. Since we were not able to interact with etGz, we hope the other reviewers and AC can verify that we have addressed etGz's concerns.

Thank you for your time,

Regards,

The Authors.

---

### Decision · Program_Chairs · 2023-01-20

**Decision:**

Accept: poster

**Justification For Why Not Higher Score:**

The surrogate methods like SP2+ does not have significant advantage against SP step. This is a weakness of the result for problems where an SP2 step does not exist.

**Justification For Why Not Lower Score:**

When SP2 step exist, the performance is impressive, and the analysis is OK.

**Metareview: Summary, Strengths And Weaknesses:**

This paper proposes an interpolation type update strategy that resembles the famous stochastic Polyak step while using second-order information. Several approaches were designed to implement this idea under different situations, including SP2, SP2+, and other versions with slack formulations.

The theoretical analysis is relatively weak since the proof is only provided for quadratic objective functions. However, this is acceptable since many popular step rules like BB and CG in practice works well, but their convergence is only available for quadratic functions. In terms of the practical performance. When SP2 step works, its performance is quite impressive in the experiments.

However, we should also notice that the SP2 step may not exist in many problems. The authors also propose the SP2+ step as the surrogate of SP2 when it doesn't exist. However, from the experiments, the SP2+ does not seem to have clear advantage compared to the existing SP step. Therefore, I think this paper is only marginally above the accept threshold.

**Note From Pc:**

if the above contains the word "oral" or "spotlight" please see: "oral" presentation means -> notable-top-5% and "spotlight" means -> notable-top-25%. As stated in our emails, we are disassociating presentation type from AC recommendations

**Summary Of Ac-Reviewer Meeting:**

N/A